# STRENGTH ESTIMATION AND HUMAN-LIKE STRENGTH ADJUSTMENT IN GAMES

**Chun-Jung Chen[1,2]\*, Chung-Chin Shih[1]\*, Ti-Rong Wu[1]†**

[1]Institute of Information Science, Academia Sinica, Taiwan
[2]Department of Computer Science, National Taiwan University, Taiwan

## ABSTRACT

Strength estimation and adjustment are crucial in designing human-AI interactions, particularly in games where AI surpasses human players. This paper introduces a novel strength system, including a *strength estimator* (SE) and an SE-based Monte Carlo tree search, denoted as *SE-MCTS*, which predicts strengths from games and offers different playing strengths with human styles. The strength estimator calculates strength scores and predicts ranks from games without direct human interaction. SE-MCTS utilizes the strength scores in a Monte Carlo tree search to adjust playing strength and style. We first conduct experiments in Go, a challenging board game with a wide range of ranks. Our strength estimator significantly achieves over 80% accuracy in predicting ranks by observing 15 games only, whereas the previous method reached 49% accuracy for 100 games. For strength adjustment, SE-MCTS successfully adjusts to designated ranks while achieving a 51.33% accuracy in aligning to human actions, outperforming a previous state-of-the-art, with only 42.56% accuracy. To demonstrate the generality of our strength system, we further apply SE and SE-MCTS to chess and obtain consistent results. These results show a promising approach to strength estimation and adjustment, enhancing human-AI interactions in games. Our code is available at https://rlg.iis.sinica.edu.tw/papers/strength-estimator.

## 1 INTRODUCTION

Artificial intelligence has achieved superhuman performance in various domains in recent years, especially in games (Silver et al., 2018; Schrittwieser et al., 2020; Vinyals et al., 2019; OpenAI et al., 2019). These achievements have raised interests within the community in exploring AI programs for human interactions, particularly in estimating human players' strengths and offering corresponding levels to increase entertainment or improve skills (Demediuk et al., 2017; Fan et al., 2019; Moon & Seo, 2020; Gusmão et al., 2015; Silva et al., 2015; Hunicke & Chapman, 2004). For example, since the advent of AlphaZero, human players have attempted to train themselves by using AI programs. Subsequently, many researchers have explored several methods to adjust the playing strength of AlphaZero-like programs to provide appropriate difficulty levels for human players (Wu et al., 2019; Liu et al., 2020; Fujita, 2022).

However, although these methods can provide strength adjustment, two issues have arisen. First, while these methods can offer different strengths, human players often need to play several games or manually choose AI playing strength, consuming time to find a suitable strength for themselves. Second, the behaviors between AI programs and human players are quite different. This occurs because most strength adjustment methods mainly focus on adjusting AI strength by calibrating the win rate to around 50% for specific strengths, without considering the human behaviors at those strengths. The problem is further exacerbated when human players attempt to use AI programs to analyze games and learn from the better actions suggested by AI. Therefore, designing AI programs that can accurately estimate a player's strength, provide corresponding playing strengths, and simultaneously offer human-like behavior is crucial for using superhuman AI in human learning.

---

\*These authors contributed equally.
†Corresponding author: tirongwu@iis.sinica.edu.tw

To address this challenge, this paper proposes a novel strength system, including a *strength estimator* and an SE-based MCTS, denoted as an *SE-MCTS*, which can predict strengths from games and provide different playing strengths with a human-like playing style. Specifically, we propose a strength estimator, based on the Bradley-Terry model (Bradley & Terry, 1952), which estimates a strength score of an action at a game state, with higher scores indicating stronger actions. The strength score can be further used to predict the strength of any given game, providing strength estimation without direct human interaction. Next, we present a novel strength adjustment approach with human-like styles, named SE-MCTS, by incorporating the strength estimator into the Monte Carlo tree search (MCTS). During the MCTS, the search is limited to exploring actions that closely correspond to a given targeted strength score. We conduct experiments in Go, a challenging board game for human players with a wide range of ranks. The results show several advantages of using our approach. First, the strength estimator significantly achieves over 80% accuracy in predicting ranks within 15 games, compared to the previous method only achieves 49% accuracy even after evaluating 100 games. Second, the experiments show that SE-MCTS can not only provide designated ranks but also achieve a playing style with 51.33% accuracy in aligning to human players' actions, while previous state-of-the-art only obtained 42.56% accuracy. Finally, the strength estimator can be trained with limited rank data and still accurately predict ranks. Furthermore, to demonstrate the generality of the proposed method, we apply SE and SE-MCTS to chess, achieving consistent results and significantly outperforming the previous state-of-the-art approach in both strength estimation and adjustment. These results show a promising direction for enhancing human-AI interactions in games.

## 2 BACKGROUND

### 2.1 BRADLEY-TERRY MODEL

The Bradley-Terry model (Bradley & Terry, 1952) is often used for pairwise comparisons, allowing for the estimation of outcomes between individuals based on their relative strengths. In a group of individuals, the model calculates the probability that individual $i$ defeats individual $j$ as $P(i \succ j) = \frac{\lambda_i}{\lambda_i + \lambda_j}$, where $\lambda_i$ and $\lambda_j$ represent the positive values of individuals $i$ and $j$, respectively. A higher $\lambda_i$ indicates a stronger individual. In practice, $\lambda_i$ is usually defined by an exponential score function as $\lambda_i = e^{\beta_i}$, where $\beta_i$ represents the strength score of individual $i$.

The Bradley-Terry model can be further generalized to include comparison among more than two individuals (Huang et al., 2006). Consider a group consisting of $k$ individuals, indexed from 1 to $k$. The probability that individual $i$ wins out over the other individuals in this group is calculated as $P(i) = \frac{\lambda_i}{\lambda_1 + \lambda_2 + ... + \lambda_k}$. Furthermore, the model can be adapted for team comparisons, where each team comprises multiple individuals. For example, assume team $a$ consists of individuals 1 and 2, team $b$ consists of individuals 2, 3, and 4, and team $c$ consists of individuals 1, 3, and 5. Then, the probability that team $a$ win against team $b$ and $c$ is defined as $P(\text{team } a) = \frac{\lambda_1 \lambda_2}{\lambda_1 \lambda_2 + \lambda_2 \lambda_3 \lambda_4 + \lambda_1 \lambda_3 \lambda_5}$, where the strength of each team is determined by the product of the strengths of its individual members. Due to its generalization and broader extension, the Bradley-Terry model has been widely used in various fields, such as games (Coulom, 2007a), sports (Cattelan et al., 2013), and recommendation systems (Chen & Joachims, 2016).

### 2.2 MONTE-CARLO TREE SEARCH

Monte Carlo tree search (MCTS) (Coulom, 2007b; Kocsis & Szepesvári, 2006) is a best-first search algorithm that has been successfully used by AlphaZero (Silver et al., 2018) and MuZero (Schrittwieser et al., 2020) to master both board games and Atari games. In AlphaZero, each MCTS simulation begins by traversing the tree from the root node to a leaf node using the PUCT (Rosin, 2011) formula:

$$a^* = \arg\max_a \left\{ Q(s,a) + c \cdot P(s,a) \cdot \frac{\sqrt{\sum_b N(s,b)}}{1 + N(s,a)} \right\}, \tag{1}$$

where $Q(s,a)$ represents the estimated Q-value for the state-action pair $(s,a)$, $N(s,a)$ is the visit counts, $P(s,a)$ is the prior heuristic value, and $c$ is a coefficient to control the exploration. Next, the leaf node is expanded and evaluated by a two-head network, $f_\theta(s) = (p, v)$, where $p$ represents the policy distribution and $v$ denotes the win rate. The policy distribution $p$ serves as the prior heuristic

value. The win rate $v$ is used to update the estimated Q-value of each node, from the leaf node to its ancestors up to the root node. This process is repeated iteratively, with more MCTS simulations leading to better decision-making. Finally, the node with the largest simulation counts is decided.

## 2.3 MCTS-BASED STRENGTH ADJUSTMENT

Strength adjustment (Hunicke & Chapman, 2004; Paulsen & Fürnkranz, 2010; Silva et al., 2015; Moon & Seo, 2020) is crucial in the design of human-AI interactions, especially since AlphaZero achieved superhuman performance in many games like Go, Chess, and Shogi. As MCTS is widely used in these games, various methods have been explored to adapt it for strength adjustment (Sephton et al., 2015; Wu et al., 2019; Demediuk et al., 2017; Fan et al., 2019; Moon et al., 2022). For instance, Sephton et al. (2015) proposes adjusting the playing strength by using a strength index $z$. After the search, MCTS decides the node based on the proportionality of their simulation counts, with the probability of selecting node $i$ calculated as $\frac{N_i^z}{\sum_{j=1}^n N_j^z}$, where $N_i$ represents the simulation counts for node $i$. A larger $z$ value indicates a tendency to select stronger actions, while a smaller $z$ favors weaker actions. Wu et al. (2019) further improves this method by introducing a threshold $R$ to filter out lower-quality actions, removing nodes $j$ where $N_j < R \times N_{max}$, where $N_{max}$ represents the node with largest simulation counts. The approach is used to adjust the playing strength of ELF OpenGo (Tian et al., 2019), resulting in covering a range of 800 Elo ratings within the interval $z \in [-2, 2]$. However, both methods only change the final decision in MCTS without modifying the search tree, leaving the search trees identical for different strengths.

## 2.4 STRENGTH ESTIMATION

Strength estimation is another important technique related to strength adjustment. With accurate strength estimation, the AI can first predict a player's strength and subsequently provide an appropriate level of difficulty for human learning. Several methods (Moudřík & Neruda, 2016; Liu et al., 2020; Egri-Nagy & Tormanen, 2020; Scheible & Schütze, 2014) have been proposed to estimate player strength in games. For example, Liu et al. (2020) proposes estimating a player's strength by using the strength index with MCTS, as described in the previous subsection, to play against human players. Specifically, the strength index $z$ is adjusted after each game according to the game outcomes. Their experiments show that $z$ generally converges after about 20 games. However, this method requires human players to play against the MCTS programs with multiple games to obtain an estimation of their playing strengths. On the other hand, Moudřík & Neruda (2016) proposes an alternative approach that categorizes the Go players into three ranks – strong, median, and weak – and uses a neural network to classify player ranks based on a game position using supervised learning. After training, given a game position, the neural network predicts ranks for each position by selecting the highest probability. Furthermore, it can aggregate predictions across multiple positions. Two methods are presented: (a) *sum*, which sums probabilities of all positions and makes a prediction based on the highest probability; and (b) *vote*, which predicts the rank of each position first and selects the most frequent rank. However, this approach does not consider multiple actions during training and the experiment was limited to only three ranks. In addition, strength estimation can be formulated as a ranking problem (Burges et al., 2005; Xia et al., 2008), but it differs in a key aspect. Ranking problems often focus on ordering items based on a single query, whereas in games, strength is assessed as overall skills across multiple positions or games. This challenge requires aggregating rankings across various scenarios to capture a player's ability.

# 3 METHOD

## 3.1 STRENGTH ESTIMATOR

We introduce the *strength estimator* (SE), which is designed to predict the strength of an action $a$ at a given state $s$ based on human game records. Each state-action pair, denoted as $p = (s, a)$, is labeled with a rank $r$ that corresponds to the player's strength. For simplicity, in this paper, ranks are ordered in descending order where rank 1, denoted as $r_1$, represents the strongest level of play, and progressively higher numbers indicate weaker playing strength. Each rank corresponds to a group of players, as we assume that players with the same rank have equivalent strength. Ranks could be

determined by Elo ratings; for example, players with an Elo between 2400 and 2599 are classified as $r_1$, players with an Elo between 2200 and 2399 as $r_2$, and so on.

Consider a game collection $\mathcal{D}$ that consists of numerous state-action pairs $p$, each associated with a distinct rank $r$. Suppose there are $n$ ranks in $\mathcal{D}$, represented by $r_1, r_2, ..., r_n$. Next, given a state-action pair $p$ sampled from $\mathcal{D}$, the strength estimator, denoted as $SE(p) = \lambda$, is designed to predict the strength $\lambda$ corresponding to action $a$ in state $s$. Consider two state-action pairs, $p_1 = (s, a_1)$ and $p_2 = (s, a_2)$, where $a_1$ and $a_2$ are played by $r_1$ and $r_2$, respectively. The strength estimator is expected to predict strength $\lambda_1$ for $p_1$ and $\lambda_2$ for $p_2$, such that $\lambda_1 \geq \lambda_2$. Note that $\lambda_1 = \lambda_2$ occurs when the actions $a_1$ and $a_2$ are identical or are of equal strength. Following the Bradley-Terry model, we can calculate the probability that $r_1$ wins against $r_2$ as $P(r_1 \succ r_2) = \frac{\lambda_1}{\lambda_1 + \lambda_2}$. To generalize to $n$ ranks, consider $n$ state-action pairs, $p_1, p_2, ..., p_n$, corresponding to ranks $r_1, r_2, ..., r_n$, respectively. The strength estimator predicts $\lambda_i$ for each $p_i$. The probability that $r_1$ wins against all other ranks is then calculated as $P(r_1 \succ \{r_2, r_3, \ldots, r_n\}) = \frac{\lambda_1}{\lambda_1 + \lambda_2 + \ldots + \lambda_n}$.

Furthermore, we extend the method to estimate the *composite strength* of a rank by incorporating multiple state-action pairs, collectively conceptualizing them as a team. This approach allows us to effectively measure the overall capabilities of players within specific ranks by considering various actions across different scenarios. Consider $m$ state-action pairs $p_{i,1}, p_{i,2}, ..., p_{i,m}$, where $p_{i,j}$ represents the $j$-th state-action pairs associated with $r_i$ sampled from $\mathcal{D}$. The strength estimator predicts the strength $\lambda_{i,1}, \lambda_{i,2}, ..., \lambda_{i,m}$ for each state-action pair, respectively. We define the composite strength for $r_i$ by aggregating all individual strengths using the geometric mean. The composite strength, denoted as $\Lambda_i$, is calculated as $\Lambda_i = (\lambda_{i,1}\lambda_{i,2}\ldots\lambda_{1,m})^{1/m}$.

The geometric mean is used to ensure that the strength estimator provides stable estimations and reflects the rank's ability across different scenarios. Namely, $\Lambda_i$ should remain consistent regardless of the number of state-action pairs considered:

$$\Lambda_i = (\lambda_{i,1}\lambda_{i,2}\ldots\lambda_{1,m})^{1/m} = \left(\prod_{j=1}^{m}\lambda_{i,j}\right)^{\frac{1}{m}} = \left(\prod_{\substack{j=1 \\ p_j \sim \mathcal{D}}}^{m} SE(p_j)\right)^{\frac{1}{m}}, \quad (2)$$

where the state-action pair $p_j$ is randomly sampled from the game collection $\mathcal{D}$. We can further calculate the probability that rank $r_1$ wins against all other ranks by using the composite strength: $P(r_1 \succ \{r_2, r_3, \ldots, r_n\}) = \frac{\Lambda_1}{\Lambda_1 + \Lambda_2 + \ldots + \Lambda_n}$.

Note that our proposed method of aggregating strength using the geometric mean for teams differs from the Bradley-Terry model, which utilizes product aggregation. However, the geometric mean is specifically tailored to our scenarios to guarantee consistent measurement of individual performance across different games, rather than focusing on team member interactions. This approach also helps to mitigate the influence of outliers. Moreover, this modification preserves the integrity of the Bradley-Terry model principles, ensuring that the order of teams is strictly followed.

## 3.2 TRAINING THE STRENGTH ESTIMATOR

This subsection introduces a methodology for training *strength estimator*. For simplicity, we propose to train a neural network, $f_\theta(p) = \beta$, as a strength estimator which predicts a strength score $\beta$ instead of strength $\lambda$ for a given state-action pair $p$. This strength score, $\beta$, serves as the exponent for strength $\lambda = e^\beta$, as defined by the Bradley-Terry model. Then, the composite strength, $\Lambda_i$, from the equation 2 can be expressed by using $\beta$ as follows:

$$\Lambda_i = \left(\prod_{j=1}^{m}\lambda_{i,j}\right)^{\frac{1}{m}} = \left(\prod_{j=1}^{m}e^{\beta_{i,j}}\right)^{\frac{1}{m}} = e^{\frac{1}{m}\sum_{j=1}^{m}\beta_{i,j}} = e^{\overline{\beta_i}}, \quad (3)$$

where $\overline{\beta_i}$ represents the average strength scores of $m$ state-action pairs, each with $r_i$, sampled from $\mathcal{D}$.

Next, given $n$ ranks in the game collection, the strength estimator is optimized by maximizing the likelihood $\mathcal{L}$ according to the ranking order (Xia et al., 2008; Chen et al., 2009). The likelihood is

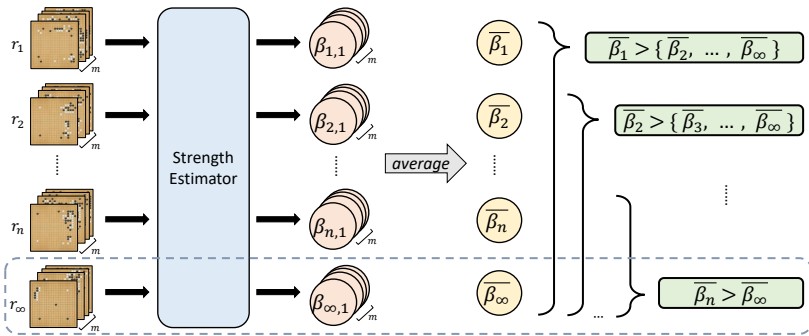

Figure 1: The training process of the strength estimator.

defined as follows:

$$\mathcal{L} = P(r_1 \succ \{r_2, r_3, \ldots, r_n\}) \times P(r_2 \succ \{r_3, r_4, \ldots, r_n\}) \times \ldots \times P(r_{n-1} \succ r_n)$$

$$= \prod_{i=1}^{n-1} P(r_i \succ \{r_{i+1}, r_{i+2}, \ldots, r_n\}) = \prod_{i=1}^{n-1} \frac{\Lambda_i}{\Lambda_i + \Lambda_{i+1} + \ldots + \Lambda_n} = \prod_{i=1}^{n-1} \frac{e^{\overline{\beta_i}}}{\sum_{j=i}^{n} e^{\overline{\beta_j}}}. \quad (4)$$

Maximizing $\mathcal{L}$ ensures that the strength scores of $r_1, r_2, \ldots, r_n$ are strictly in descending order, such that $r_1 \succ r_2 \succ r_3 \succ \ldots r_n$. Then, the loss function $L$ can be defined by log-likelihood:

$$L = -\log(\mathcal{L}) = -\log\left(\prod_{i=1}^{n-1} \frac{e^{\overline{\beta_i}}}{\sum_{j=i}^{n} e^{\overline{\beta_j}}}\right) = -\sum_{i=1}^{n-1} \log\left(\frac{e^{\overline{\beta_i}}}{\sum_{j=i}^{n} e^{\overline{\beta_j}}}\right). \quad (5)$$

Figure 1 illustrates the training process of the strength estimator. Assume there are $n$ ranks, $r_1, r_2, \ldots, r_n$, in the game collection. Initially, for each $r_i$, we sample $m$ state-action pairs and evaluate them by the strength estimator. The strength estimator then outputs the strength score $\beta_{i,j}$ corresponding to each state-action pair. Subsequently, we average all $\beta_{i,j}$ to obtain $\overline{\beta_i}$ for each $r_i$. Finally, using all strength scores, $\overline{\beta_i}$, we sequentially minimize each softmax loss as defined by the equation 5.

Since the state-action pairs are collected only from human games, the strength estimator may provide unpredictable estimations for out-of-distribution state-action pairs, which rarely appear in human games. To address this issue, we introduce an additional rank, $r_\infty$ into our training process, as depicted by the dashed rectangle in Figure 1. This rank, $r_\infty$, is defined as the weakest among all ranks, ensuring that $r_i \succ r_\infty$ for all $r_i$. To generate the state-action pairs for $r_\infty$, we first select a state-action pair, $p_i = (s_i, a_i)$ from any $r_i$. Then, we disturb the state-action pair $p_i$ to $p_\infty$ by modifying the action $a_i$ to a randomly chosen legal action, resulting in $p_\infty = (s_i, a_\infty)$. Since a random action $a_\infty$ is highly likely to result in an inferior outcome, these actions are expected to correspond to the weakest rank. Note that although a random action may occasionally result in a strong action, the impact of such outliers will be minimized by the average $\overline{\beta_\infty}$ as the number of samples, $m$, increases.

### 3.3 STRENGTH ESTIMATOR BASED MCTS FOR STRENGTH ADJUSTMENT

We present a novel method that integrates a strength estimator with MCTS to adjust strength dynamically, named *SE-MCTS*. In previous strength adjustment approaches, as described in subsection 2.3, the MCTS search tree is unmodified during the search, with only changing the final action decision after the search is complete. In contrast, we propose inherently modifying the search based on a target strength score to ensure that the search aligns more closely with the desired strength of ranks.

Specifically, in MCTS each node is evaluated by the strength estimator to obtain a strength score, $\beta(s, a)$, which represents the strength score of action $a$ at state $s$. We can calculate the composite strength score $\overline{\beta}(s, a)$, by averaging all $\beta$ from the nodes within the subtree of state $s$. This is similar to the method used to calculate estimated Q-values. Given a targeted rank $r$ with strength score $\beta_t$, we

calculate the absolute strength difference for each node, which is denoted as $\delta(s,a) = |\overline{\beta}(s,a) - \beta_t|$. Higher values of $\delta$ indicate that the action is unlikely chosen by a player of the target rank, while lower $\delta$ suggest that the actions are closer to the strength of the target rank.

As the values $\delta(s,a)$ are unbounded and can be any non-negative number, we normalize all values using the minimum and maximum values observed in the current search tree. This normalization ensures that the difference values are bounded within the $[0,1]$ interval, similar to the approach used in MuZero (Schrittwieser et al., 2020). Then, the PUCT formula in MCTS selection is modified from equation 1 as follows:

$$a^* = \arg\max_a \left\{ Q(s,a) + c \cdot \left( P(s,a) - c_1 \cdot \hat{\delta}(s,a) \right) \cdot \frac{\sqrt{\sum_b N(s,b)}}{1 + N(s,a)} \right\}, \qquad (6)$$

where $\hat{\delta}(s,a)$ is the normalized difference values of $\delta(s,a)$ and $c_1$ is a hyperparameter used to control the confidence of prior heuristic values and difference values. Note that we choose to use $\hat{\delta}(s,a)$ to eliminate the prior heuristic value $P(s,a)$, rather than incorporating additional values like the use of $Q(s,a)$. This is because, during the MCTS search, the algorithm first prioritizes actions based on higher prior heuristic values and gradually shifts the focus to actions with higher Q-values when the simulation counts increase. By combining $P(s,a)$ and $\hat{\delta}(s,a)$, we effectively adjust the prioritization of actions, thereby aligning the search more closely with the desired strengths.

## 4 EXPERIMENTS

### 4.1 EXPERIMENT SETUP

We conducted experiments using the MiniZero framework (Wu et al., 2025). The human games are collected[1] from FoxWeiqi (Foxwq, 2025), which is the largest online Go platform in terms of users. These games are collected from amateur 5 *kyu* to 9 *dan*[2], and are ranked in order from the strongest to weakest as follows: 9 dan, 8 dan, ..., 2 dan, 1 dan, 1-2 kyu, and 3-5 kyu, corresponding to $r_1$, $r_2$, ..., $r_{11}$. Namely, a total of $n = 11$ ranks are used. Note that for kyu, we classify 1 to 2 kyu as one rank and 3 to 5 kyu as another rank, and we exclude games played by players ranked lower than 5 kyu. This is because kyu players are still mastering basic Go strategies, their ranks often change rapidly. Consequently, their games do not consistently correspond to their ranks. For the training dataset, we collect a total of 495,000 games, with 45,000 games from each rank. We also prepare a separate testing dataset, including a *candidate* and a *query* dataset. The candidate dataset is used to estimate an average strength score of each rank, including a total of 1,100 games, with 100 games per rank. The query dataset is used for the strength estimator to predict the strength, containing a total of 9,900 games, with 900 games per rank.

The network architecture of the strength estimator is similar to the AlphaZero network, consisting of 20 residual blocks with 256 channels. Given a state-action pair, the network outputs a policy distribution $p$, a value $v$, and a strength score $\beta$. The training loss for the policy and value network follows AlphaZero, while the loss for the strength estimator is defined by equation 5. During training, we aggregate the composite strength score $\overline{\beta_i}$ by randomly selecting $m = 7$ state-action pairs from $r_i$. Other training details are provided in the appendix.

### 4.2 PREDICTING RANKS FROM GAMES

The strength estimator can be utilized to predict ranks in games where the rank is unknown. We first calculate $\overline{\beta_i}$ for each $r_i$ by evaluating all games in the candidate dataset. Next, for games from the same unknown rank, $r_u$, in the query dataset, a composite score $\overline{\beta_u}$ is calculated by the strength estimator. Finally, $r_u$ is then determined to be $r_i$, where $|\overline{\beta_u} - \overline{\beta_i}|$ is the smallest among all $r_i$.

We train two strength estimator networks, SE and $\text{SE}_\infty$, where SE is trained with 11 ranks, and $\text{SE}_\infty$ includes an additional rank, $r_\infty$, for a total of 12 ranks. In addition, for comparison, we train another

---

[1]We downloaded Go games from the FoxWeiqi online platform using its public download links.

[2]In the game of Go, *Kyu* represents the beginner to decent amateur level, ranging from 18 kyu to 1 kyu, with lower numbers indicating stronger kyu players; *Dan* denotes advanced amateur, ranging from 1 dan to 9 dan, with higher numbers indicating stronger dan players.

network based on supervised learning (SL), $\texttt{SL}_{\text{sum}}$ and $\texttt{SL}_{\text{vote}}$, as mentioned in subsection 2.4. Both $\texttt{SL}_{\text{sum}}$ and $\texttt{SL}_{\text{vote}}$ are trained to classify 11 ranks for a given state-action pair but with different aggregation methods.

The evaluation is conducted as follows. For each $r_i$ from the query dataset, each network evaluates all state-action pairs from $N$ randomly selected games and then predicts a rank. We repeat this prediction process 500 times for each $N$ to ensure a stable estimation.

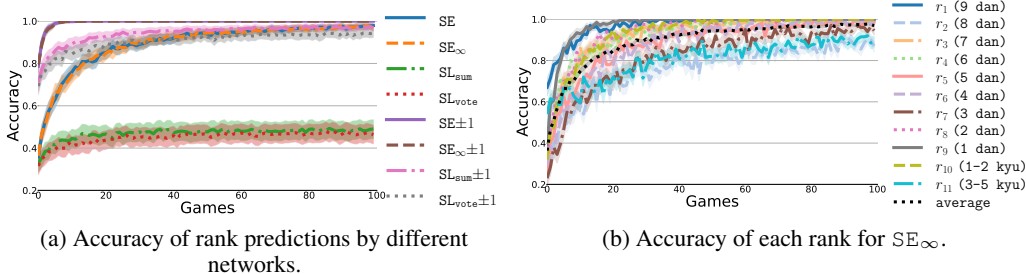

(a) Accuracy of rank predictions by different networks.

(b) Accuracy of each rank for $\texttt{SE}_\infty$.

Figure 2: Accuracy of rank prediction in Go, with the shaded area representing the 95% confidence interval.

Figure 2a shows the accuracy of predicting the games from the query dataset. From the figure, the two strength estimator networks significantly outperform the supervised learning networks. Both $\texttt{SE}$ and $\texttt{SE}_\infty$ perform nearly identical performance, achieving over 80% of accuracy with only 15 games and reaching an accuracy of 97.5% after evaluating 100 games. In contrast, both $\texttt{SL}_{\text{sum}}$ and $\texttt{SL}_{\text{vote}}$ reached an accuracy of 49%, even after evaluating 100 games, with $\texttt{SL}_{\text{sum}}$ performs slightly better than $\texttt{SL}_{\text{vote}}$. Furthermore, as human players do not always perform consistently and may occasionally change their ranks by one rank either above or below their actual rank, the games within $r_i$ might involve players whose actual ranks are $r_{i-1}$ or $r_{i+1}$, leading to slight fluctuations in the dataset. Therefore, we incorporate a prediction tolerance that allows for a deviation of one rank. Specifically, if the network predicts $r_{i-1}$ or $r_{i+1}$ for $r_i$, we consider this prediction accurate. The results show that both strength estimator networks achieve nearly 80% accuracy by only evaluating a single game, and perfectly predict the rank with an accuracy of over 99% after only 6 games, while the supervised learning networks still cannot predict the correct ranks after 100 games. This result indicates that the ranks predicted by the strength estimators are close to the actual ranks even when the prediction is incorrect.

Figure 2b depicts the accuracy of each rank prediction for $\texttt{SE}_\infty$ with several interesting observations. First, $r_1$ (9 dan) has the highest accuracy among all ranks. Since 9 dan is the highest rank on FoxWeiqi, so professional Go players and Go AIs are also classified at this level, thus creating a significant strength gap between $r_1$ and $r_2$, and leading to a clearer distinction. This phenomenon results in some players, originally ranked at $r_1$ (9 dan), who are relatively stronger compared to $r_2$ (8 dan), being relegated to $r_2$. Consequently, $r_2$ shows the lowest accuracy among all ranks. Second, the prediction for $r_7$ (3 dan) is below the average. This is because new players on FoxWeiqi can only select a maximum initial rank of 3 dan and must advance gradually. Therefore, many players at this rank are actually stronger than 3 dan. Finally, the prediction for $r_{11}$ (3-5 kyu) is also below average, corroborating to common understanding that the strength of kyu players is usually inconsistent. In conclusion, these results allow our strength estimator to provide evaluations of a game's ranking system, further offering developers to make adjustments. Details on the accuracy of each rank prediction for other networks are provided in the appendix.

## 4.3 ADJUSTING STRENGTH WITH STRENGTH ESTIMATOR

In this section, we evaluate the performance of SE-MCTS, as described in subsection 3.3, by incorporating the two trained strength estimator networks into MCTS to adjust the playing strength for game playing. We first calculate the composite strength score $\overline{\beta_i}$, by averaging all $\beta$ from the state-action pairs in $r_i$ from the candidate set. Although we assume that the strength score $\beta_i$ for any state-action pair from $r_i$ should be similar, in practice, we observe that $\overline{\beta_i}$ may vary across different

action numbers, as shown in Figure 3. In the beginning, particularly for the first actions, all $\overline{\beta_i}$ are estimated as the same score. This is likely because the action choices at the beginning of Go have less variety, and weaker players can easily remember and imitate the openings from stronger players. Therefore, in the SE-MCTS, we propose using $\overline{\beta_i^d}$ instead of $\overline{\beta_i}$ as the target strength score for each action, where $\overline{\beta_i^d}$ represent the composite strength score at $d$-th action for $r_i$.

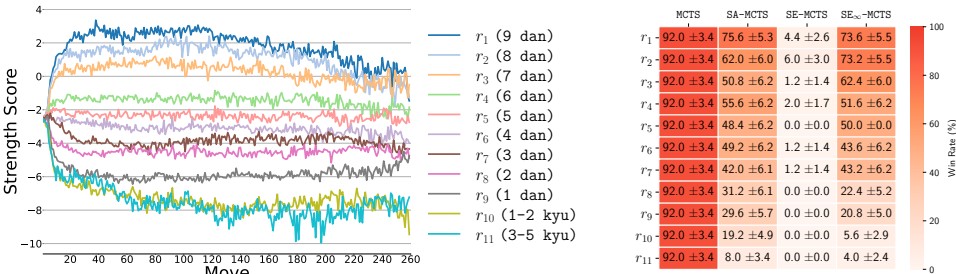

| Figure 3: The composite strength score from $SE_\infty$ for each rank across different actions in games. | Figure 4: Win rate against $SE_\infty$-$MCTS_5$ in Go. |
|---|---|

To evaluate the performance, we select four MCTS programs: (a) `MCTS`, representing vanilla MCTS without any strength adjustment mechanism, (b) `SA-MCTS`, which utilizes the strength index $z$ (Wu et al., 2019), (c) `SE-MCTS`, which uses `SE` network, and (d) `SE∞-MCTS`, which uses `SE∞` network. Except `MCTS`, the remaining three programs can be adjusted to different strengths across a total of 11 ranks. We use `SA-MCTS`$_i$, `SE-MCTS`$_i$, and `SE∞-MCTS`$_i$ to represent the strength of each program correspond to $r_i$. For `SE-MCTS` and `SE∞-MCTS`, we calculate the target strength score from the candidate dataset for strength adjustment. However, for `SA-MCTS`, since the strength index $z$ does not directly correspond to any specific rank, we adjust $z$ for each $r_i$ to ensure that each `SA-MCTS`$_i$ and `SE∞-MCTS`$_i$ achieve a comparable win rate, i.e., approximately 50%. Each $z$ is in Appendix C.

Figure 4 shows the win rate for each program playing against a baseline program, where the baseline is chosen as `SE∞-MCTS`$_i$ with $i = 5$ (5 dan). Generally, the win rate for `SA-MCTS`$_i$ decreases as $i$ increases, except from $i = 3$ to $i = 6$ where there is a slight fluctuation. This corroborates the experiments in (Wu et al., 2019). Interestingly, although `SE` can accurately predict the strength, `SE-MCTS`$_i$ cannot adjust the strength effectively. This is due to the exploration in the MCTS search, which may inevitably explore actions that are rarely seen in human games. Without training using $r_\infty$, `SE` provides inaccurate strength scores for these unseen actions. In contrast, the win rate of `SE∞-MCTS`$_i$ consistently decreases as $i$ increases, demonstrating an accurate strength adjustment using strength scores. We also include an experiment with different baselines in Appendix C.1.

Moreover, we are interested in whether these programs can align with the human player's behavior, i.e., if they can choose the same actions as human players. Therefore, we sample 50 human games from the query dataset for each rank and use four MCTS programs to play on every state-action pair. The accuracy is evaluated based on whether the programs choose the same actions as human players. The results are shown in Table 1. From the table, generally, the accuracy of $r_i$ decreases when the number $i$ increases for all four programs. This is because weaker players are more unpredictable

Table 1: Accuracy of move prediction for MCTS programs to human players in Go.

| $rank\ (dan/kyu)$ | MCTS | SA-MCTS | SE-MCTS | SE∞-MCTS |
|---|---|---|---|---|
| $r_1$ (9 dan) | 53.05% ± 0.95% | 47.00% ± 0.95% | 53.06% ± 0.95% | **53.73% ± 0.95%** |
| $r_2$ (8 dan) | 53.79% ± 0.97% | 45.83% ± 0.97% | 53.96% ± 0.97% | **54.30% ± 0.97%** |
| $r_3$ (7 dan) | 52.70% ± 0.98% | 46.70% ± 0.98% | **54.28% ± 0.97%** | 53.88% ± 0.98% |
| $r_4$ (6 dan) | 52.50% ± 0.92% | 45.86% ± 0.92% | **54.25% ± 0.92%** | 53.58% ± 0.92% |
| $r_5$ (5 dan) | 49.48% ± 0.93% | 42.29% ± 0.92% | **52.00% ± 0.93%** | 50.35% ± 0.93% |
| $r_6$ (4 dan) | 49.44% ± 0.91% | 42.72% ± 0.90% | **53.11% ± 0.90%** | 50.87% ± 0.91% |
| $r_7$ (3 dan) | 50.75% ± 0.89% | 42.68% ± 0.88% | **53.71% ± 0.89%** | 51.40% ± 0.89% |
| $r_8$ (2 dan) | 50.17% ± 0.93% | 40.94% ± 0.92% | **53.21% ± 0.93%** | 50.99% ± 0.93% |
| $r_9$ (1 dan) | 48.10% ± 0.89% | 40.94% ± 0.88% | **52.60% ± 0.89%** | 49.44% ± 0.89% |
| $r_{10}$ (1-2 kyu) | 46.95% ± 0.91% | 36.58% ± 0.88% | **50.06% ± 0.91%** | 47.84% ± 0.91% |
| $r_{11}$ (3-5 kyu) | 46.87% ± 0.89% | 36.64% ± 0.86% | **51.36% ± 0.89%** | 48.23% ± 0.89% |
| average | 50.35% ± 0.28% | 42.56% ± 0.28% | **52.87% ± 0.28%** | 51.33% ± 0.28% |

than stronger players. For `MCTS`, it achieves a high accuracy, exceeding 50%, but it cannot adjust strength as shown in Figure 4. In contrast, although `SA-MCTS` can adjust strengths, it achieves the lowest accuracy among all four programs. This is due to `SA-MCTS` selecting actions proportional to the simulation counts by using strength index $z$. The randomness results in diverging from human playing styles. On the other hand, both `SE-MCTS` and $\text{SE}_\infty\text{-MCTS}$ achieve a high accuracy, even better than `MCTS`. This is because SE-MCTS directly modifies the search by the strength scores, guiding the search to better align with human behavior. In conclusion, among all four programs, $\text{SE}_\infty\text{-MCTS}$ not only adjusts strengths to specific ranks but also provides playing styles that are closely aligned with those of human players at specific ranks.

## 4.4 TRAINING STRENGTH ESTIMATOR WITH LIMITED DATA

In this subsection, we investigate training a strength estimator with limited data. Unlike the supervised learning methods, $\text{SL}_{\text{sum}}$ and $\text{SL}_{\text{vote}}$, which require data from each rank, our method can estimate a strength score and use it to predict ranks that were not observed during training. In niche games or those favored by a specific group of enthusiasts, ranking systems are often not fully established due to a limited number of game records, and some specific ranks may be sparsely populated with only a few players. Therefore, it is intriguing to explore whether the strength estimator can generalize to these unseen strengths.

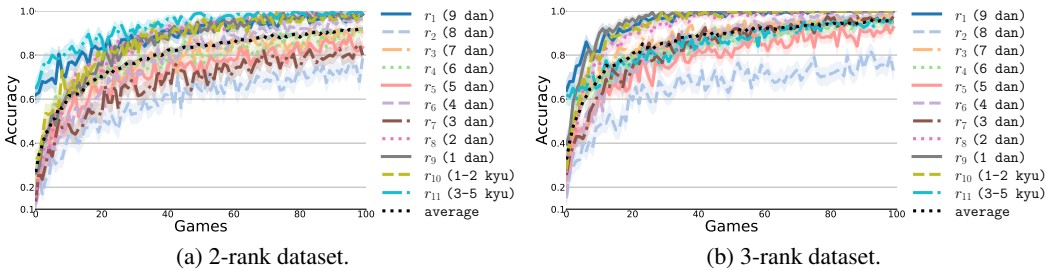

(a) 2-rank dataset.  (b) 3-rank dataset.

Figure 5: Accuracy of rank predictions on the limited dataset of Go.

We train $\text{SE}_\infty$ on two separate datasets: the 2-rank dataset, containing $r_1$ (9 dan) and $r_{11}$ (3-5 kyu), and the 3-rank dataset, which includes the same ranks as the 2-rank dataset, plus an additional rank, $r_6$ (4 dan). During the evaluation, we utilize the same methods, as described in 4.2, by using the strength estimator to calculate $\overline{\beta_i}$ for each $r_i$ in both candidate and query datasets, and then perform the predictions. Figure 5a and 5b show the accuracy of rank prediction on 2-rank and 3-rank datasets, respectively. On average, the strength estimator achieves an accuracy of over 80% after evaluating 38 games for the 2-rank dataset and 21 games for the 3-rank datasets, respectively. Although these numbers are larger than 15 – the number of games needed by a strength estimator trained with all 11 ranks to achieve over 80% accuracy – it can still effectively predict ranks that were not seen during training. This suggests that the strength estimator can generalize across a spectrum of ranks with few ranks. Note that if sufficient data for more ranks are available, the accuracy still improves. Our method provides a way to both generalize with limited data and enhance performance as more data becomes available.

## 4.5 GENERALIZING TO OTHER GAMES

We further experiment in another game, chess, to demonstrate the generality of our SE and SE-MCTS approaches. Similar to Go, chess is also a popular game with abundant human game records. The games were collected from Lichess[3] (Lichess, 2025), which uses Elo ratings as its ranking system. We collect games with Elo ratings ranging from 1,000 to 2,600 and categorize them into eight ranks, with each rank covering 200 Elo points and 240,000 games, for a total of 1,920,000 games. For the testing dataset, the candidate dataset consists of 960 games, with 120 games per rank, while the query dataset contains 9,600 games, with 1,200 games per rank. Then, we apply experiments to chess,

---

[3]Lichess is one of the most popular online chess platforms, with millions of active users.

similar to those in Go. Note that the training algorithms remain identical, except the input features of the neural network changed from Go to chess.

The results are consistent with the findings in Go. First, as shown in Figure 6, $SE_\infty$ achieves over 80% accuracy in predicting ranks with only 26 games, significantly outperforming both $SL_{sum}$ and $SL_{vote}$, and reaches an accuracy of 93.38% after evaluating 100 games. Second, Figure 7 shows the win rate for each $SE_\infty\text{-}MCTS_i$ when playing against $SE_\infty\text{-}MCTS_5$. The win rate consistently decreases as $i$ increases, demonstrating $SE_\infty\text{-}MCTS$ can adjust its strength in chess. Finally, Table 2 shows the accuracy of predicting human moves, with $SE_\infty\text{-}MCTS$ achieving an average of 47.25% accuracy in aligning with human actions, outperforming both $MCTS$ (46.56%), $SA\text{-}MCTS$ (40.21%), and $SE\text{-}MCTS$ (38.93%). In conclusion, this experiment demonstrates the versatility of our approach.

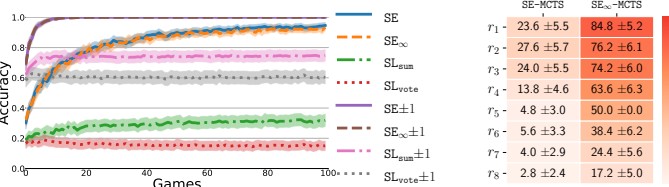

| Figure 6: Accuracy of rank prediction in chess. | Figure 7: Win rate against $SE_\infty\text{-}MCTS_5$ in chess. |
|---|---|

Table 2: Accuracy of move prediction for MCTS programs to human chess players.

| $rank$ ($Elo$) | MCTS | SA-MCTS | SE-MCTS | $SE_\infty$-MCTS |
|---|---|---|---|---|
| $r_1$ (2400 − 2599) | **51.97% ± 0.69%** | 50.17% ± 0.69% | 41.05% ± 0.68% | 51.51% ± 0.69% |
| $r_2$ (2200 − 2399) | **51.58% ± 0.69%** | 47.49% ± 0.69% | 40.19% ± 0.68% | 51.14% ± 0.69% |
| $r_3$ (2000 − 2199) | **49.23% ± 0.69%** | 45.01% ± 0.69% | 41.40% ± 0.68% | 49.19% ± 0.69% |
| $r_4$ (1800 − 1999) | 46.52% ± 0.69% | 41.78% ± 0.68% | 38.66% ± 0.67% | **47.26% ± 0.69%** |
| $r_5$ (1600 − 1799) | 45.45% ± 0.69% | 38.18% ± 0.67% | 36.76% ± 0.67% | **46.62% ± 0.69%** |
| $r_6$ (1400 − 1599) | 44.33% ± 0.69% | 36.84% ± 0.67% | 37.63% ± 0.67% | **46.12% ± 0.69%** |
| $r_7$ (1200 − 1399) | 41.54% ± 0.68% | 31.49% ± 0.64% | 37.65% ± 0.67% | **43.04% ± 0.68%** |
| $r_8$ (1000 − 1199) | 41.89% ± 0.68% | 30.72% ± 0.64% | 38.05% ± 0.67% | **43.08% ± 0.68%** |
| average | 46.56% ± 0.24% | 40.21% ± 0.24% | 38.93% ± 0.24% | **47.25% ± 0.24%** |

## 5 DISCUSSION

This paper introduces a novel strength system, including a *strength estimator* for evaluating the strength from game records without requiring direct interaction with human players, and an *SE-MCTS* for adjusting the playing strength using strength scores provided by the strength estimator. When predicting ranks in the game of Go, our strength estimator significantly achieves over 80% accuracy by examining only 15 games, whereas the previous supervised learning method only reached 49% accuracy even after evaluating 100 games. The strength estimator can be trained with limited rank data and still accurately predict unseen rank data, providing extensive generalizability. For strength adjustment, SE-MCTS successfully adjusts to designated ranks while providing a playing style that aligns with human behavior, achieving an average accuracy of 51.33%, compared to the previous state-of-the-art method that only reached 42.56% accuracy. Furthermore, we apply our method to the game of chess and obtain consistent results to Go, demonstrating the generality of our approach.

One limitation of our work is that the strength estimator relies on human game records for training. However, this issue could potentially be addressed by using all models trained by AlphaZero, which may serve as players of different playing strengths to generate games. Besides, the strength system also provides several benefits for future directions. For example, game designers can use the strength estimator to evaluate their ranking systems. The strength estimator can evaluate a game by examining the strength scores for each action, and use it to identify incorrect actions for human players or for cheat detection (Alayed et al., 2013). Furthermore, we can extend our strength estimator by incorporating opponent-specific strength scores to address the Bradley-Terry model's limitations in capturing intransitivity (Balduzzi et al., 2018; Bertrand et al., 2023; Omidshafiei et al., 2019; Vadori & Savani, 2024). Finally, the search tree of SE-MCTS can offer the opportunity for explainability of AI actions in human learning.

ETHICS STATEMENT

This paper presents a method for estimating player strength and adjusting it to specific ranks, allowing agents to play in a human-like manner. A potential ethical concern is that our method could be exploited by human players to develop human-like agents for cheating in human competitions. However, we emphasize that all models trained in this paper are used strictly for research purposes and adhere to established ethical guidelines.

REPRODUCIBILITY STATEMENT

We have provided detailed descriptions of the method, implementation, and training hyperparameters in Section 3, Section 4, and Appendix A to facilitate the reproduction of our experiments. The source code, along with a README file containing instructions is available at https://rlg.iis.sinica.edu.tw/papers/strength-estimator.

ACKNOWLEDGEMENT

This research is partially supported by the National Science and Technology Council (NSTC) of the Republic of China (Taiwan) under Grant Number NSTC 113-2221-E-001-009-MY3, NSTC 113-2634-F-A49-004, and NSTC 113-2221-E-A49-127.

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

## A    Detailed Training Settings for the Strength Estimator

The feature design in the strength estimator of Go is similar to AlphaZero (Silver et al., 2018). Specifically, we use 18 channels to represent a board position, where the first 16 channels are the board configurations from the past eight moves for both black and white stones. The remaining two channels are binary indicators of the color of the next player, i.e., one channel for Black and White. For chess, the feature design also follows the same approach as AlphaZero, which includes 119 input channels. During training, for each rank, we randomly select seven state-action pairs. We also perform data augmentation to further enhance the diversity of the training data. The network is optimized using stochastic gradient descent (SGD), with the loss function specified in Equation 5. It is important to note that when training $SE_\infty$, the policy and value loss for state-action pairs of $r_\infty$ are not calculated, since these heads should only consider actual human players' actions. The learning rate is initially set at 0.01 and is halved after 100,000 training steps. The entire training process encompasses 130,000 steps, consuming around 242 GPU hours for Go and 69 GPU hours for chess on an NVIDIA RTX A5000 graphics card. Other hyperparameters are listed in Table 3.

Table 3: Hyperparameters for training strength estimators.

| Parameter | Go | Chess |
|---|---|---|
| Number of Blocks | 20 | 20 |
| Input Channel | 18 | 119 |
| Hidden Channel | 256 | 256 |
| Learning Rate | 0.01 to 0.005 | 0.01 to 0.005 |
| Training Steps | 130,000 | 130,000 |
| Optimizer | SGD | SGD |
| Main Memory | 384GB | |
| Central Processing Unit (CPU) | Intel Xeon Silver 4216 (2.1 GHz) | |
| Graphical Processing Unit (GPU) | NVIDIA RTX A5000 | |
| GPU Hours | 242 | 69 |

## B    In-depth Analysis for Strength Estimator in Go

We conduct in-depth analyses for strength estimators in Go. First, we present detailed insights into predicting ranks from games, as detailed in Subsection B.1. Second, we demonstrate the outcomes of strength prediction using fewer moves in a single game, discussed in Subsection B.2. Finally, we explore predictions based solely on the first 50 actions or the last 50 actions in games, which are elaborated in Subsection B.3 and Subsection B.4, respectively.

### B.1    Predicting Ranks from Games

Figure 8 shows the accuracy of rank predictions for different networks. We observe that in Figure 8a and Figure 8b, $SL_{vote}$ and $SL_{sum}$ can only distinguish on some ranks, such as 3-5 kyu. This is because these models do not contain sufficient information to differentiate all ranks based on a single state-action during training (Moudřík & Neruda, 2016). In Figure 8c and Figure 8d, even though we incorporated a prediction tolerance for these two methods, they still cannot perfectly distinguish all ranks, even after 100 games. In Figure 8e and Figure 8f, although our models $SE$ and $SE_\infty$ cannot perfectly predict all ranks without incorporating a prediction tolerance, they still achieve high performance across all ranks. In Figure 8g and Figure 8h, when we allow a prediction tolerance, we achieve 100% accuracy across all ranks. This result further indicates that our model can differentiate the strength relationship across all ranks.

### B.2    Predicting Ranks from Game Positions

We are also interested in whether we can predict the rank using only game positions. Specifically, only one game position can be chosen for each game instead of all actions when predicting the

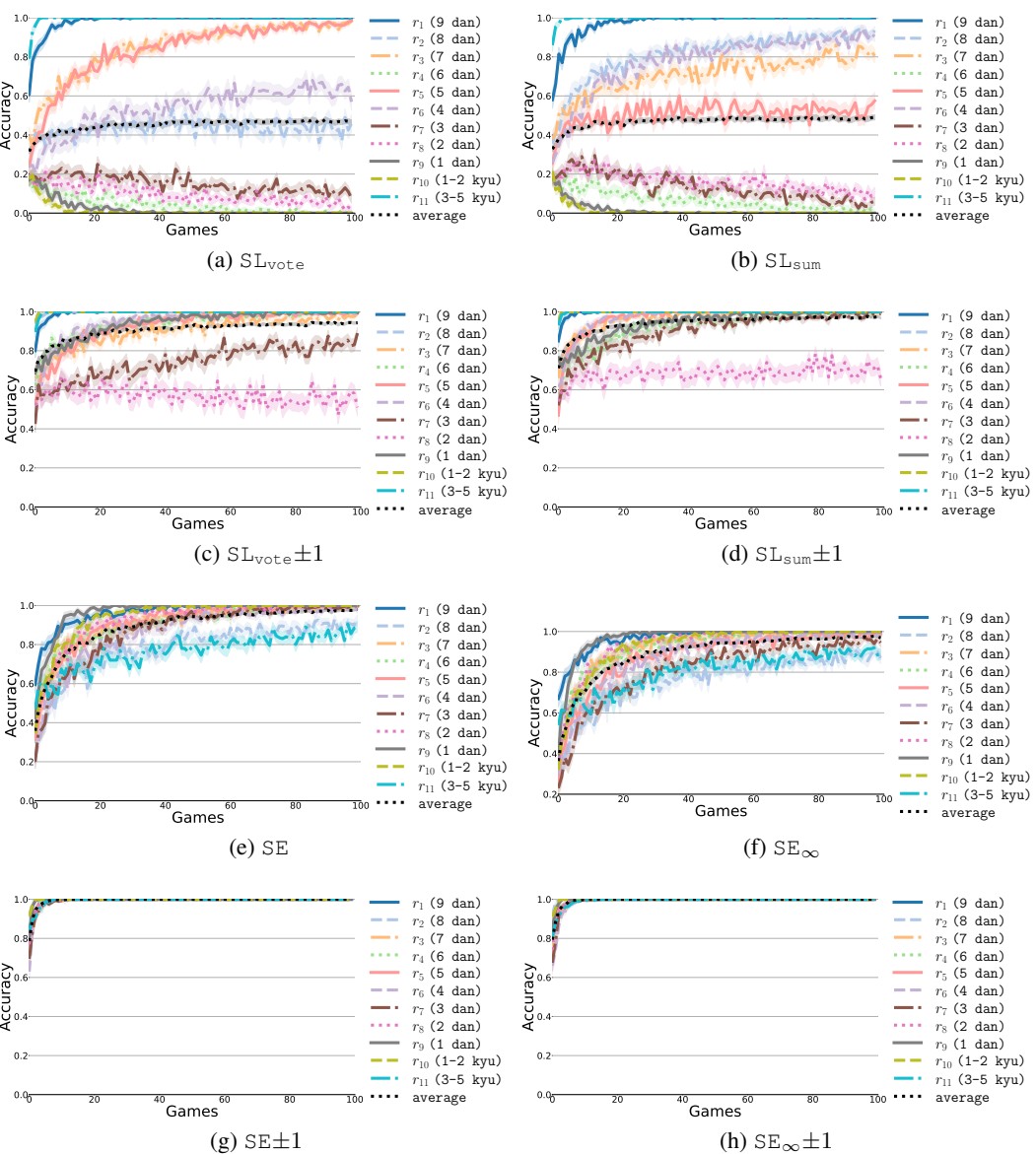

Figure 8: Accuracy of rank prediction for different networks in Go.

ranks, shown in Figure 9. According to Figures 9a and 9b, both $SL_{vote}$ and $SL_{sum}$ show similar performance as in the situation of using all actions. In our method, Figures 9c and 9d demonstrate that across 20 games, the accuracy decreases from 80%, when predictions are based on all actions, to approximately 60% when using just one action. This is intuitive because using the information from the entire game would help capture the player's strength. However, achieving 60% accuracy with unrelated actions among 20 games indicates that our model can still predict accurate ranks based on one action of different games.

## B.3 PREDICTING RANKS FROM THE FIRST 50 ACTIONS IN THE GAME

We are interested in evaluating performance when using only the first 50 actions of a game, known as the *fuseki* stage in Go. Figures 10a and 10b indicate that $SL_{vote}$ and $SL_{sum}$, utilizing these initial actions, achieve similar performance to predictions made using all actions in the game. Figures 10c and 10d present the prediction results by our methods when limited to the first 50 actions.

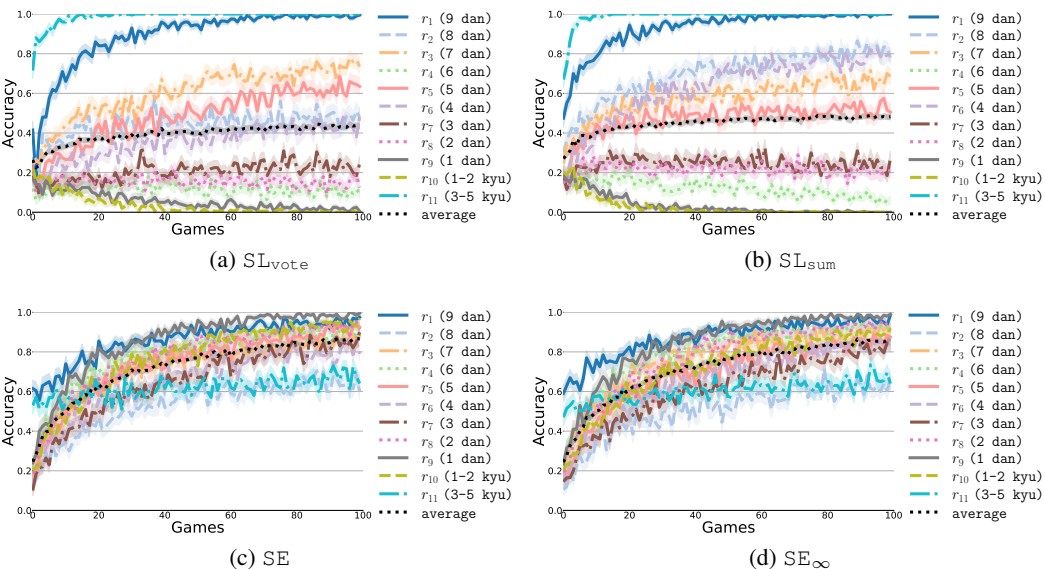

Figure 9: Accuracy of rank prediction for different networks in the game positions of Go.

Clearly, the overall accuracy has declined. Notably, the accuracy for kyu players shows significant downward trends. This is mainly because the actions in the *fuseki* stage at these ranks are similar, thus complicating accurate predictions. This suggests that enhancing performance during the *fuseki* could be crucial for human players aiming to progress from kyu to dan rank.

### B.4 PREDICTING RANKS FROM THE LAST 50 ACTIONS IN THE GAME

Similarly, we examine the performance when using only the last 50 actions of the game, referred to as the *yose* stage in Go. Figures 10e and 10f show that $SL_{vote}$ and $SL_{sum}$, employing these final actions, maintain similar performance to predictions based on all actions in the game. In our method, Figures 10g and 10h display the prediction results using only the last 50 actions. As before, the overall accuracy has declined. However, the accuracy for 9 dan players between predictions made using the entire game and just the last 50 actions does not differ significantly. This is likely because the *yose* stage involves complex calculations and judgments, areas where top players excel. Furthermore, in most games, especially those between the highest-skilled players, the outcome is often determined before the *yose* stage. This leads to less practice and proficiency in this phase among players of lower ranks. Additionally, we observe a significant drop in accuracy for 8 dan players when predictions are based solely on *yose* stage. This could be because some 8 dan players have comparable *yose* skills to those of 9 dan players, leading to some misclassifications of 8 dan players as 9 dan.

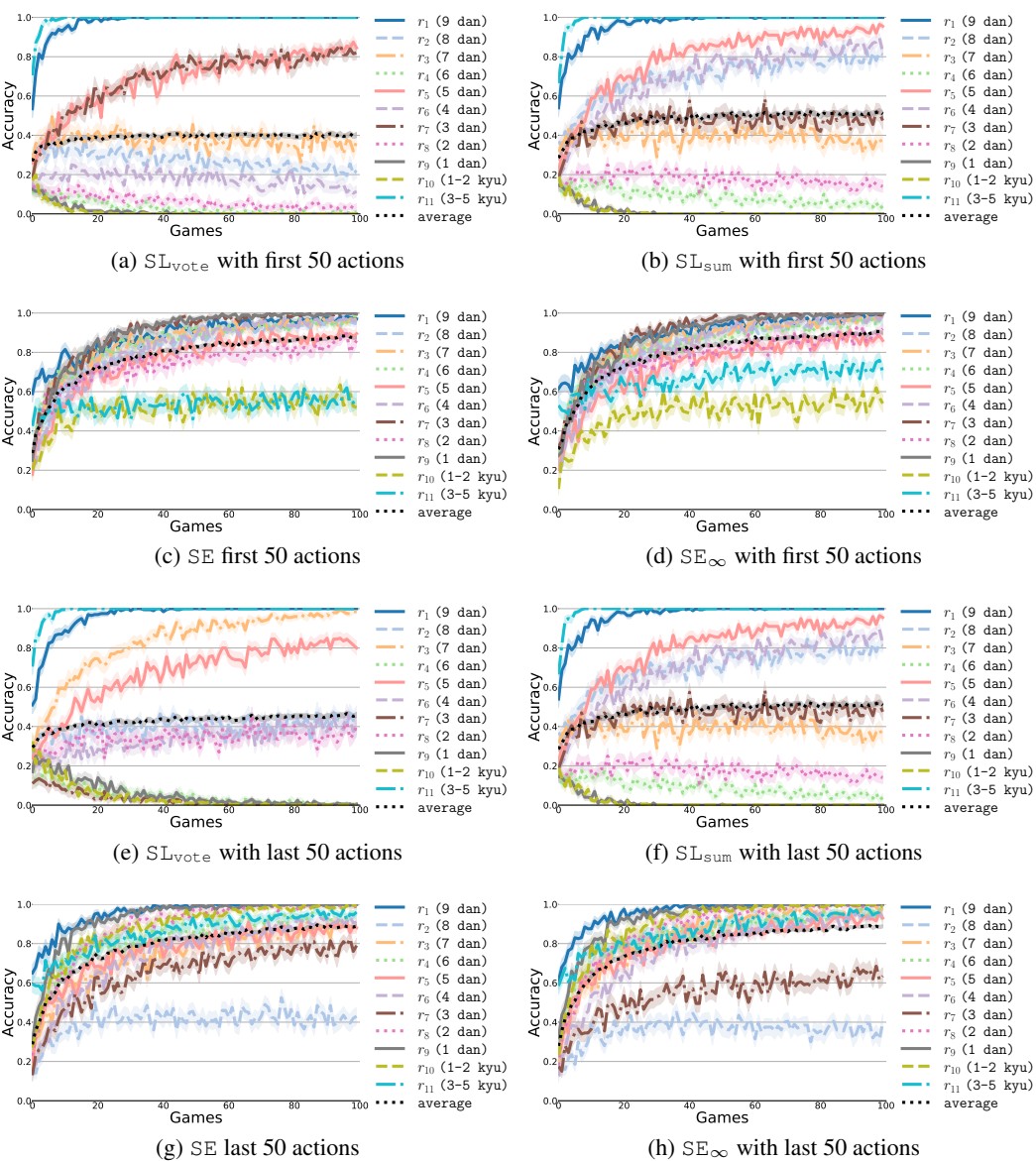

Figure 10: Accuracy of rank prediction for different networks using only the first or the last 50 actions of the game in Go.

## C    DETAILED EXPERIMENTS FOR STRENGTH ADJUSTMENT

Table 4 presents the value of $z$ for $\mathtt{SA-MCTS}_i$ used in subsection 4.3. To ensure that each $\mathtt{SA-MCTS}_i$ and $\mathtt{SE}_\infty\mathtt{-MCTS}_i$ achieve a comparable win rate, all methods are tested using a fixed simulation count of 800. For $\mathtt{SA-MCTS}$, since the strength index $z$ does not directly correspond to any specific rank, we adjust $z$ for each $r_i$ to ensure that each $\mathtt{SA-MCTS}_i$ and $\mathtt{SE}_\infty\mathtt{-MCTS}_i$ achieve a comparable win rate. As shown in Table 4, $z$ gradually decreases from $r_1$ to $r_{11}$, aligning with the results in the original paper, which indicate that a greater $z$ corresponds to a higher strength.

Table 4: $z$ according to rank.

| Rank | $z$ |
|---|---|
| $r_1$ (9 dan) | 0.6 |
| $r_2$ (8 dan) | 0.5 |
| $r_3$ (7 dan) | 0.35 |
| $r_4$ (6 dan) | 0.3 |
| $r_5$ (5 dan) | 0.2 |
| $r_6$ (4 dan) | 0.15 |
| $r_7$ (3 dan) | 0.05 |
| $r_8$ (2 dan) | -0.1 |
| $r_9$ (1 dan) | -0.2 |
| $r_{10}$ (1-2 kyu) | -0.6 |
| $r_{11}$ (3-5 kyu) | -1.0 |

### C.1    ADJUSTING STRENGTH WITH DIFFERENT BASELINES

In Figure 4, the baseline program is chosen as $\mathtt{SE}_\infty\mathtt{-MCTS}_i$ with $i = 5$ (5 dan). It would be interesting to examine whether the relative strength remains consistent when different baseline models are used. To further investigate this, we conduct a round-robin tournament by selecting five ranks ($r_2$, $r_4$, $r_6$, $r_8$ and $r_{10}$) and two representative methods ($\mathtt{SA-MCTS}$ and $\mathtt{SE}_\infty\mathtt{-MCTS}$, excluding $\mathtt{SE-MCTS}$ due to its ineffective strength adjustment. Each combination involves 250 games, requiring approximately 100 GPU hours on an NVIDIA RTX A5000. Table 5 summarizes the results, with the win rates in each cell representing the performance of the y-axis player against the x-axis player. Moreover, we compute the Elo rating of each model using this table. We initialize the rating at 1500 for each model and iteratively update the ratings to match the expected win rates with the observed pairwise outcomes. The rightmost column of Table 5 presents the resulting Elo ratings. In summary, the Elo ratings confirm that higher-ranked models consistently achieve higher ratings, demonstrating the robustness of our method across different baselines.

Table 5: The round-robin tournament among ten MCTS programs: $\mathtt{SA-MCTS}_2$, $\mathtt{SA-MCTS}_4$, $\mathtt{SA-MCTS}_6$, $\mathtt{SA-MCTS}_8$, $\mathtt{SA-MCTS}_{10}$, $\mathtt{SE}_\infty\mathtt{-MCTS}_2$, $\mathtt{SE}_\infty\mathtt{-MCTS}_4$, $\mathtt{SE}_\infty\mathtt{-MCTS}_6$, $\mathtt{SE}_\infty\mathtt{-MCTS}_8$, $\mathtt{SE}_\infty\mathtt{-MCTS}_{10}$. For simplicity, abbreviations $\mathtt{SA}\text{-}r_2$, $\mathtt{SA}\text{-}r_4$, etc. are used.

| | SA-$r_2$ | SA-$r_4$ | SA-$r_6$ | SA-$r_8$ | SA-$r_{10}$ | SE$_\infty$-$r_2$ | SE$_\infty$-$r_4$ | SE$_\infty$-$r_6$ | SE$_\infty$-$r_8$ | SE$_\infty$-$r_{10}$ | Avg. Win Rate | Elo |
|---|---|---|---|---|---|---|---|---|---|---|---|---|
| SA-$r_2$ | - | 64.4% ± 5.95% | 65.2% ± 5.92% | 77.6% ± 5.18% | 90.8% ± 3.59% | 56.0% ± 6.17% | 64.8% ± 5.93% | 67.2% ± 5.83% | 76.0% ± 5.30% | 94.0% ± 2.95% | 72.9% ± 1.8% | 1685.62 |
| SA-$r_4$ | 35.6% ± 5.95% | - | 58.4% ± 6.12% | 67.2% ± 5.83% | 85.2% ± 4.41% | 37.6% ± 6.02% | 55.6% ± 6.17% | 62.0% ± 6.03% | 75.2% ± 5.36% | 86.0% ± 4.31% | 62.5% ± 2.0% | 1605.22 |
| SA-$r_6$ | 34.8% ± 5.92% | 41.6% ± 6.12% | - | 65.6% ± 5.90% | 84.4% ± 4.51% | 35.6% ± 5.95% | 40.4% ± 6.09% | 50.0% ± 6.21% | 66.0% ± 5.88% | 80.0% ± 4.97% | 55.4% ± 2.1% | 1550.24 |
| SA-$r_8$ | 22.4% ± 5.18% | 32.8% ± 5.83% | 34.4% ± 5.90% | - | 68.0% ± 5.79% | 24.8% ± 5.36% | 39.2% ± 6.06% | 39.2% ± 6.06% | 47.6% ± 6.20% | 68.8% ± 5.75% | 41.9% ± 2.0% | 1452.23 |
| SA-$r_{10}$ | 9.2% ± 3.59% | 14.8% ± 4.41% | 15.6% ± 4.51% | 32.0% ± 5.79% | - | 7.2% ± 3.21% | 14.8% ± 4.41% | 21.6% ± 5.11% | 22.8% ± 5.21% | 50.8% ± 6.21% | 21.0% ± 1.7% | 1279.80 |
| SE$_\infty$-$r_2$ | 44.0% ± 6.17% | 62.4% ± 6.02% | 64.4% ± 5.95% | 75.2% ± 5.36% | 92.8% ± 3.21% | - | 70.4% ± 5.67% | 83.2% ± 4.64% | 88.0% ± 4.04% | 93.6% ± 3.04% | 74.9% ± 1.8% | 1706.28 |
| SE$_\infty$-$r_4$ | 35.2% ± 5.93% | 44.4% ± 6.17% | 59.6% ± 6.09% | 60.8% ± 6.06% | 85.2% ± 4.41% | 29.6% ± 5.67% | - | 61.2% ± 6.05% | 78.0% ± 5.15% | 85.6% ± 4.36% | 60.0% ± 2.0% | 1586.26 |
| SE$_\infty$-$r_6$ | 32.8% ± 5.83% | 38.0% ± 6.03% | 50.0% ± 6.21% | 60.8% ± 6.06% | 78.4% ± 5.11% | 16.8% ± 4.64% | 38.8% ± 6.05% | - | 72.0% ± 5.58% | 80.0% ± 4.97% | 52.0% ± 2.1% | 1526.51 |
| SE$_\infty$-$r_8$ | 24.0% ± 5.30% | 24.8% ± 5.36% | 34.0% ± 5.88% | 52.4% ± 6.20% | 77.2% ± 5.21% | 12.0% ± 4.04% | 22.0% ± 5.15% | 28.0% ± 5.58% | - | 69.6% ± 5.71% | 38.2% ± 2.0% | 1423.82 |
| SE$_\infty$-$r_{10}$ | 6.0% ± 2.95% | 14.0% ± 4.31% | 20.0% ± 4.97% | 31.2% ± 5.75% | 49.2% ± 6.21% | 6.4% ± 3.04% | 14.4% ± 4.36% | 20.0% ± 4.97% | 30.4% ± 5.71% | - | 21.3% ± 1.7% | 1283.25 |

