# OpenReview forum: "Strength Estimation and Human-Like Strength Adjustment in Games"
_ICLR.cc/2025/Conference — ICLR 2025 Poster_

### Official Review · Reviewer_651t · 2024-10-31

**Soundness:** 3
**Presentation:** 4
**Contribution:** 3
**Rating:** 6
**Confidence:** 4

**Summary:**

This paper introduces a dedicated strength estimator and an SE-based Monte Carlo tree search,  which predicts strengths from games
and offers different playing strengths with human styles. A large amount of expert data with different playing level is used for training, and the learned strength is empoyed to adjust the prior policy in MCTS. Starting from the Bradley-Terry model, a new loss function is proposed. The experimental results demonstrate that the model can effectively predict the level of actions and guide MCTS in making decisions at various levels in both Go and Chess.

**Strengths:**

1. This paper is well-written with a clear and easily understandable structure.

2. This paper is highly motivated and the novelty is good.

3. The experiments are comprehensive, with cross-validation conducted using two applications.

**Weaknesses:**

1. SE-MCTS adjusts the prior probabilities by the strength level $\hat{\delta}$  to influence the decision-making level of MCTS, and the number of simulations has not been reduced. It seems that SA-MCTS can give actions with $k$ different strength level by setting different $z$ for just one search process, but SE-MCTS need $k$ searches for $k$ different strength level actions. Besides aligning more closely with the desired strength rank, what are the more advantages of modifying the search process compared with performing on the final action decision?

2. SE-MCTS achieves different levels of actions by modifying prior probabilities. However, as mentioned in the paper, with an increase in the number of simulations, the decision-making gradually shifts towards relying on Q-values. Does this imply that with a sufficient number of simulations, SE-MCTS might become ineffective, generatinig the same action with different $\hat{\delta}$? Can you provide a ablation study of SE-MCTS under different numbers of MCTS simulations?

3.  In Figure 4, why the win rate of SE-MCTS so low?

4. Because different players have different strategies, Player A's defeat Player B does not necessarily indicate that A is stronger than B. It may simply mean that A's policy happened to exploit a weakness in B's policy. You can have pairwise matches between SE-MCTS, SA-MCTS, and MCTS with different levels, then rank them using Elo rating system, which would be more convincing.

**Questions:**

Please refer to the weaknesses.

---

> ### Author Response · Authors · 2024-11-19
> **Response to Reviewer 651t**
>
> We appreciate the thoughtful comments from the reviewers and address your questions in detail below.
>
> > … what are the more advantages of modifying the search process compared with performing on the final action decision?
>
> The key advantage of SE-MCTS over SA-MCTS is its potential for enhanced human learning and explainability. First, as the reviewer noted, SE-MCTS better aligns with the desired strength rank, which is crucial for real-world applications like learning systems for Go or chess players. Second, as SE-MCTS modifies the search tree itself, it aligns more closely with the thought process of the desired rank, opening opportunities for improved explainability to help players understand and improve their decision-making. In contrast, while SA-MCTS allows a single search to produce actions at multiple strength levels, it leads to inconsistencies – such as weak players performing advanced moves due to the randomness – resulting in unrealistic playing styles.
>
> > … with a sufficient number of simulations, SE-MCTS might become ineffective … provide a ablation study of SE-MCTS under different numbers of MCTS simulations?
>
> Yes, this phenomenon is common in all MCTS-based programs, including SA-MCTS, due to the inherent trade-off between exploration and exploitation in UCB. As the number of simulations increases, MCTS prioritizes exploitation, causing move selections to converge. As the reviewer suggested, we conducted experiments using 800 (same as the paper), 1200, and 1600 simulations. The results below show that while accuracy slightly decreases for all MCTS-based programs as the number of simulations increases, SE-MCTS consistently outperforms MCTS and SA-MCTS, demonstrating its robustness.
>
> ||MCTS|SA-MCTS|SE-MCTS|SE\$_{\infty}$-MCTS|
> |-|-|-|-|-|
> |Average (sim=800)|50.35%±0.28%|42.56%±0.28%|52.87%±0.28%|51.33%±0.28%|
> |Average (sim=1200)|49.30%±0.28%|42.40%±0.28%|52.55%±0.28%|50.09%±0.28%|
> |Average (sim=1600)|48.05%±0.28%|41.69%±0.28%|52.74%±0.28%|49.53%±0.28%|
>
> >In Figure 4, why the win rate of SE-MCTS so low?
>
> We have addressed this issue in the third paragraph of subsection 4.2 in our initial version. This arises because MCTS inherently explores low-probability moves – those less likely to be selected by human players – due to its exploration mechanism. Without incorporating an additional $r_{\infty}$ rank during training, SE-MCTS struggles to handle these rarely seen actions. If the explanation in subsection 4.2 is insufficient, we welcome the reviewer's feedback for further clarification.
>
> > … have pairwise matches between SE-MCTS, SA-MCTS, and MCTS with different levels, then rank them using Elo rating system …
>
> Running the pairwise matches like in Figure 4 is highly resource-demanding, as each match includes 250 games and requires approximately 100 GPU hours on an NVIDIA RTX A5000.
>
> We attempt to conduct a smaller round-robin experiment during the rebuttal period. Due to the computational challenges, we selected three ranks ($r_4$, $r_6$, and $r_8$) and two representative methods (SA-MCTS and SE$_\infty$-MCTS), excluding SE-MCTS due to its ineffective strength adjustment. Each combination involves 250 games. The results below demonstrate our method's robustness across different baselines. The win rates in each cell are from the perspective of the y-axis player playing against the x-axis player. To calculate the Elo rating of each model, we initialize the rating at 1500 and iteratively update them to align the expected win rates with the observed pairwise win rates. The rightmost column of the table presents the coverage Elo ratings.
>
> ||SA-MCTS$_4$|SA-MCTS$_6$|SA-MCTS$_8$|SE$_\infty$-MCTS$_4$|SE$_\infty$-MCTS$_6$|SE$_\infty$-MCTS$_8$|avg. win rate|Elo|
> |-|-|-|-|-|-|-|-|-|
> |**SA-MCTS$_4$**|-|58.4%±6.12%|67.2%±5.83%|55.6%±6.17%|62.0%±6.03%|75.2%±5.36%|63.7%±2.64%|1587.13|
> |**SA-MCTS$_6$**|41.6%±6.12%|-|65.6%±5.9%|40.4%+-6.09%|50.0%±6.21%|66.0%±5.88%|54.2%±2.71%|1518.81|
> |**SA-MCTS$_8$**|32.8%±5.83%|34.4%±5.9%|-|39.2%±6.06%|39.2%±6.06%|47.6%±6.2%|38.6%±2.69%|1432.55|
> |**SE$_\infty$-MCTS$_4$**|44.4%±6.17%|59.6%+-6.09%|60.8%±6.06%|-|61.2%±6.05%|78.0%±5.15%|55.4%±2.66%|1569.51|
> |**SE$_\infty$-MCTS$_6$**|38.0%±6.03%|50.0%±6.21%|60.8%±6.06%|38.8%±6.05%|-|72.0%±5.58%|47.9%±2.68%|1515.01|
> |**SE$_\infty$-MCTS$_8$**|24.8%±5.36%|34.0%±5.88%|52.4%±6.2%|22.0%±5.15%|28.0%±5.58%|-|32.2%±2.61%|1391.83|
>
> If the reviewer finds this table helpful, we are open to extending the round-robin table with more models and including the table in the appendix of our final version.
>
> ---
>
> Thank you again for your valuable reviews. We are open to further discussions or conducting additional experiments.

---

> ### Author Response · Authors · 2024-11-25
> **Follow Up**
>
> Dear Reviewer 651t,
>
> Thank you for your review and valuable feedback. As the discussion phase is ending, we would greatly appreciate it if you could review our rebuttal.
>
> To further strengthen our empirical results, we have included a round-robin tournament and rank the players based on Elo rating system in the Appendix. We believe these additional experiments provide more convincing results and hope the reviewer will consider a more favorable recommendation.
>
> We are more than willing to provide additional explanations or experiments within the remaining discussion period. Thank you again for your time and consideration.
>
> Sincerely,
>
> Authors

---

### Official Review · Reviewer_KBHz · 2024-11-03

**Soundness:** 2
**Presentation:** 3
**Contribution:** 3
**Rating:** 6
**Confidence:** 3

**Summary:**

This paper introduces a strength estimator to estimate the strength of GO and chess players. Additionally, a modification of the MCTS algorithm based on the strength estimator is introduced, to make it possible to adjust the playing strength of the MCTS agent to certain levels. The data for training the strength estimator is collected from FoxWeiqi, the largest online GO platform, and models are trained to estimate the gameplay levels from 3-5kyu to 9 dan, which covers most of the meaningful strength levels of human players. The experiments show the strength estimator significantly outperforms traditional supervised learning approaches, and the gameplay strength of $\text{SE}_\infty$-MCTS agent appears to be correlated to the target strength.

**Strengths:**

This paper presents the proposed methods clearly with sufficient details. The proposed methods are described with proper motivations. According to experiment results, the proposed strength estimator shows significant improvement compared to the traditional supervised learning algorithms. The results also show the proposed method can adjust the playing strength to some extent.

**Weaknesses:**

To my understanding, the strength estimator method is relevant to learning to rank [1], [2]. I believe there is some novelty, but some related background could be discussed to make the novelty more clear.

[1] Burges, Chris, et al. "Learning to rank using gradient descent." Proceedings of the 22nd international conference on Machine learning. 2005.

[2] Li, Hang. "A short introduction to learning to rank." IEICE TRANSACTIONS on Information and Systems 94.10 (2011): 1854-1862.

I believe there are some ways to improve the clarity and soundness of experiment parts. For Figure 4, I would like to recommend authors expand each column to a 2D map (except for MCTS). That means comparing SA-MCTS, SE-MCTS, and $\text{SE}_\infty$-MCTS for all combinations of ranks, instead of comparing all ranks with only $r_5$. I understand there is a page limit but such figures can be included in the appendix.

Some more concerns about the experiment are listed in my Questions and can be addressed by updating the paper.

**Questions:**

1.	Is it possible to train and test SE-MCTS on Chess like on GO? I believe no matter whether the results are good or bad, adding such results will always strengthen the paper.
2.	What is the exact value of $z$ for each rank?
3.	Which strength estimator is used in Figure 3?
4.	What does the strength score of vanilla MCTS and SA-MCTS look like? I am interested in the strength score curves (Figure 3) of vanilla MCTS and SA-MCTS (with varied strength index/rank settings).
5.	Which paper is the reference for SA-MCTS? I understand it is referred to in subsection 2.3, but there are multiple citations in 2.3, and am not sure which one proposed SA-MCTS.

---

> ### Author Response · Authors · 2024-11-19
> **Response to Reviewer KBHz (1/2)**
>
> We appreciate the constructive feedback and have conducted additional experiments to provide detailed answers to reviewer's questions below.
>
> > … the strength estimator method is relevant to learning to rank [1], [2] … some related background could be discussed to make the novelty more clear.
>
> Strength estimation is similar to learning to rank but differs in key aspects. To clarify the distinction between strength estimation and learning to rank, the revision includes a discussion at the end of subsection 2.4, as follows:
>
> "In addition, strength estimation is similar to ranking problems (Burges et al., 2005; Xia et al., 2008), but it differs in a key aspect. Ranking problems often focus on ordering items based on a single query, whereas in games, strength is assessed as overall skills across multiple positions or games. This challenge requires aggregating rankings across various scenarios to capture a player's ability."
>
> Thank you for pointing this out.
>
> > … expand each column to a 2D map (except for MCTS). That means comparing SA-MCTS, SE-MCTS, and SE∞-MCTS for all combinations of ranks …
>
> We understand that a round-robin table would provide more comprehensive results. However, this experiment is highly resource-demanding. In Figure 4, each cell includes 250 games to ensure statistically robust results, following the SA-MCTS paper. Running each cell requires approximately 100 GPU hours on an NVIDIA RTX A5000.
>
> A full round-robin table, as suggested by the reviewer, comparing SE-MCTS, SA-MCTS, and SE$_{\infty}$-MCTS would involve 528 combinations (11 ranks * 3 methods * 32 / 2 matches), requiring 52,800 GPU hours or 2,200 GPU days, which is impractical. Therefore, we simply follow the SA-MCTS approach, comparing all variants against a baseline to reduce computational overhead.
>
> Nevertheless, we attempt to conduct a smaller round-robin experiment during the rebuttal period. Due to the computational challenges, we selected three ranks ($r_4$, $r_6$, and $r_8$) and two representative methods (SA-MCTS and SE$_\infty$-MCTS), excluding SE-MCTS due to its ineffective strength adjustment. Each combination involves 250 games. The results below demonstrate our method's robustness across different baselines. The win rates in each cell are from the perspective of the y-axis player playing against the x-axis player. To calculate the Elo rating of each model, we initialize the rating at 1500 and iteratively update them to align the expected win rates with the observed pairwise win rates. The rightmost column of the table presents the coverage Elo ratings.
>
> ||SA-MCTS$_4$|SA-MCTS$_6$|SA-MCTS$_8$|SE$_\infty$-MCTS$_4$|SE$_\infty$-MCTS$_6$|SE$_\infty$-MCTS$_8$|avg. win rate|Elo|
> |-|-|-|-|-|-|-|-|-|
> |**SA-MCTS$_4$**|-|58.4%±6.12%|67.2%±5.83%|55.6%±6.17%|62.0%±6.03%|75.2%±5.36%|63.7%±2.64%|1587.13|
> |**SA-MCTS$_6$**|41.6%±6.12%|-|65.6%±5.9%|40.4%+-6.09%|50.0%±6.21%|66.0%±5.88%|54.2%±2.71%|1518.81|
> |**SA-MCTS$_8$**|32.8%±5.83%|34.4%±5.9%|-|39.2%±6.06%|39.2%±6.06%|47.6%±6.2%|38.6%±2.69%|1432.55|
> |**SE$_\infty$-MCTS$_4$**|44.4%±6.17%|59.6%+-6.09%|60.8%±6.06%|-|61.2%±6.05%|78.0%±5.15%|55.4%±2.66%|1569.51|
> |**SE$_\infty$-MCTS$_6$**|38.0%±6.03%|50.0%±6.21%|60.8%±6.06%|38.8%±6.05%|-|72.0%±5.58%|47.9%±2.68%|1515.01|
> |**SE$_\infty$-MCTS$_8$**|24.8%±5.36%|34.0%±5.88%|52.4%±6.2%|22.0%±5.15%|28.0%±5.58%|-|32.2%±2.61%|1391.83|
>
> If the reviewer finds this table helpful, we are open to extending the round-robin table with more models and including the table in the appendix of our final version.

---

> ### Author Response · Authors · 2024-11-19
> **Response to Reviewer KBHz (2/2)**
>
> > Is it possible to train and test SE-MCTS on Chess like on GO?
>
> We initially excluded SE-MCTS on chess because it performs worse than SE$_{\infty}$-MCTS in the Go experiment. However, it is possible to train and test SE-MCTS on chess. We have updated both Figure 6 and Table 2 to include the results for SE-MCTS. For Figure 7, the experiments are still in progress due to the higher computational demands, as explained in the previous question. We will update the revision as soon as these experiments are completed.
>
> > What is the exact value of z for each rank?
>
> The exact values of $z$  for each rank are listed below.
> |Rank|z|
> |-|-|
> |$r_1$ (9dan)|0.6|
> |$r_2$ (8dan)|0.5|
> |$r_3$ (7dan)|0.35|
> |$r_4$ (6dan)|0.3|
> |$r_5$ (5dan)|0.2|
> |$r_6$ (4dan)|0.15|
> |$r_7$ (3dan)|0.05|
> |$r_8$ (2dan)|-0.1|
> |$r_9$ (1dan)|-0.2|
> |$r_{10}$ (1-2kyu)|-0.6|
> |$r_{11}$ (3-5kyu)|-1|
>
> > Which strength estimator is used in Figure 3?
>
> The strength estimator used in Figure 3 is SE$_{\infty}$. We have revised the caption of Figure 3.
>
> > What does the strength score of vanilla MCTS and SA-MCTS look like?
>
> To conduct this experiment, we run 100 self-play games for both vanilla MCTS and SA-MCTS at each rank. The strength scores for each model are listed below. For vanilla MCTS, its strength score is around 1, corresponding to a rank close to $r_1$. Interestingly, for SA-MCTS, the strength score looks arbitrary. This is because SA-MCTS incorporates random action selection after the MCTS search, leading to inconsistent performance (sometimes playing very weak moves and other times very strong moves). This further corroborates that SA-MCTS is less human-like and less conducive to human learning.
>
> |method|score|
> |-|-|
> |MCTS|1.018488|
> |SA-MCTS$_1$|-0.60355|
> |SA-MCTS$_2$|-0.65297|
> |SA-MCTS$_3$|-1.05996|
> |SA-MCTS$_4$|-1.58343|
> |SA-MCTS$_5$|-0.91812|
> |SA-MCTS$_6$|-1.36419|
> |SA-MCTS$_7$|-0.76572|
> |SA-MCTS$_8$|-1.67664|
> |SA-MCTS$_9$|-1.04419|
> |SA-MCTS$_{10}$|-1.56062|
> |SA-MCTS$_{11}$|-1.54722|
>
> > Which paper is the reference for SA-MCTS?
>
> We apologize for the confusion. SA-MCTS was proposed by Wu et al. (2019). We have added the citation to the second paragraph of subsection 4.3, where SA-MCTS is first introduced.
>
> ---
>
> We hope the additional experiments and revisions improve the clarity and soundness of our work. We are open to further discussions or conducting additional experiments.

---

> > ### Comment · Reviewer_KBHz · 2024-11-23
> >
> > Thank you for your response. I believe the round-robin table is helpful and most of my questions and concerns are addressed. I think the table of exact values of $z$ for each rank could be added to the appendix and I hope the authors could briefly describe how did they determine these values.
> >
> > I believe the core idea of this paper is valuable for the game AI community and I am convinced that the authors will make all efforts to improve the comprehensiveness of the experiment part (e.g., extending the round-robin table). Therefore, I am pleased to raise my score.

---

> > > ### Author Response · Authors · 2024-11-24
> > > **Response to Reviewer KBHz**
> > >
> > > Thank you very much for raising the score. We truly appreciate your recognition of our work and your support!
> > >
> > > We have uploaded a new revision and included the strength estimator for chess in Figure 7 and the following in Appendix C:
> > > - A table of exact values of $z$ and how these values were determined.
> > > - A round-robin table as provided in our previous response.
> > >
> > > Please let us know if you have any additional suggestions. We hope this revision contributes to a more favorable recommendation for our paper.

---

### Official Review · Reviewer_jMjA · 2024-11-03

**Soundness:** 2
**Presentation:** 3
**Contribution:** 2
**Rating:** 3
**Confidence:** 4

**Summary:**

This paper derives a strength estimator network from human game-plays labeled with rating data and then use the SE network to inform MCTS such that the policy plays at a specific rank in a human-like fashion. The strength estimator is learned with a loss function derived from a Bradley-Terry model.

**Strengths:**

I find the application of a SE network applied to adjusting the strength of game playing agents an interesting application with practical utility. The loss function derived from the Bradley-Terry model for learning the SE network is novel and with generalisation to more than two player settings.

**Weaknesses:**

I have several concerns with the proposed approach in this paper. First, the Bradley-Terry model represents each candidates with a scalar score (e.g. Elo score), which has many well-documented limitations [1-4]. A salient limitation is its inability to capture intransitivity, which would become a limitation factor for the SE network as it assumes that game plays by a player at rank $r' > r$ would win against game play by a player at rank $r$ when the lower ranked play may have played an effective exploiter strategy. Perhaps the scope of the method should be limited to perfect-information games? Second, the authors have not discussed the issue of reach-probability of states in defining the loss function to the SE network. For a state-action pair (s, a), the output of SE(s, a) is most heavily influenced by the player rank that most often visit state $s$, yet this does not seem to be accounted for in the method presented. While the issue of state coverage came up, it's not clear that adding randomised exploration policies would offset the influence of different state reach probability. Finally, the notion of "strength" features heavily in writing, however, I find it difficult to pin down exactly what "strength" means as it should depend on both the player and its opponent's strategies. The distribution over strategies seem implicit and dependent on the empirical game play data.

[1] https://arxiv.org/abs/2206.12301

[2] https://arxiv.org/abs/1806.02643

[3] https://arxiv.org/abs/1903.01373

[4] https://arxiv.org/abs/2306.05366

**Questions:**

1. L53: "..., with higher scores indicating stronger actions": could you clarify what stronger implies here? Is it actions that lead to expected win-rates? If so, what is the expectation over?
2. L58: "... correspond to a given targeted strength score": am I right in thinking that the quality of the strength adjustment of SE-MCTS in state s relies heavily on the quality of SE(s, a)? If so, could it be the case that players at a certain rank would rarely reach state s in the first place and lead to inaccurate prediction by SE-MCTS?
3. L173: "... aggregating using the geometric mean": it would help readers if the use of the geometric mean can be better motivated. Is it because of how it works out in Eq (3) for comparing a rank to multiple other ranks? It could be helpful to state this explicitly.
4. L238: "we sequentially perform the softmax loss...", typo? minimise the softmax loss?
5. an intuitive baseline, given that the authors have access to ranked game play, would be to optimise a rank-conditioned policy network $\pi(\cdot | s, r)$ with $r$ the set of player ranks. If I understood correctly, methods such as SA-MCTS does not require ranked game-play data whereas the proposed methods do.

---

> ### Author Response · Authors · 2024-11-19
> **Response to Reviewer jMjA (1/2)**
>
> We thank the reviewer for providing valuable comments and address concerns as follows.
>
> > … A salient limitation is its inability to capture intransitivity … game plays by a player at rank r′>r would win against game play by a player at rank r when the lower ranked play may have played an effective exploiter strategy …
>
> We would like to clarify that **strength represents a player's overall skill or ranking within a group**. For instance, on online Go platforms like FoxWeiqi, a player ranked as 2 dan reflects a higher overall strength (or win rate) compared to other 1 dan players. However, a 2 dan player may not always achieve a higher win rate against every 1 dan player due to exploiter strategies. Based on this foundation, a player with a higher strength score indicates a higher ranking or Elo rating and is highly likely (though not always guaranteed) to achieve a higher win rate against players with lower strength scores.
>
> We understand the limitations of the Bradley-Terry model, particularly its inability to account for intransitivity. However, the Bradley-Terry model remains widely used in games and sports such as Go, chess, and football due to its simplicity and reliable strength estimation. Our experiments focus on predicting a player's ranking as it would appear on online platforms, and we do not claim to address individual transitivity cases, as it is beyond the scope of this paper. We recognize this as an interesting challenge for future research and have revised our discussion accordingly.
>
> > … exactly what "strength" means as it should depend on both the player and its opponent's strategies …
>
> > L53: "..., with higher scores indicating stronger actions": … what stronger implies here? …
>
> As mentioned in the previous question, "strength" represents a player's overall skill or ranking within a group. Each player's strength score is independent of opponent strategies. Regarding *higher scores indicating stronger actions*, selecting actions with higher strength scores leads to a higher expected ranking compared to those with lower strength scores.
>
> > … the issue of reach-probability of states in defining the loss function to the SE network …
>
> > L58: … the quality of the strength adjustment of SE-MCTS in state s relies heavily on the quality of SE(s, a)? … players at a certain rank would rarely reach state s in the first place and lead to inaccurate prediction by SE-MCTS?
>
> We kindly request clarification on the "issue of reach-probability of states in defining the loss function", as we are uncertain of its meaning. If this refers to whether all possible states can be examined during training, the answer is no. It is well-known that, when training neural networks on human games, many states rarely appear or are never reached. This phenomenon is not unique to the SE network – it also affects policy and value networks. This is why AlphaGo relies on neural network generalizability to extrapolate from seen to unseen states. Regarding whether states rarely visited by players at certain ranks could lead to inaccurate predictions by SE-MCTS: yes, this can occur, as it can with MCTS and SA-MCTS. However, we do not view this phenomenon as a significant issue.
>
> We apologize if our explanation does not fully address the reviewer's concern and would appreciate further clarification for a more comprehensive discussion.
>
> > L173: "... aggregating using the geometric mean": it would help readers if the use of the geometric mean can be better motivated …
>
> We kindly remind the reviewer that we have explained the reasoning for using geometric mean in the fourth paragraph in subsection 3.1. Specifically, we explain that the geometric mean ensures stable estimations and reflects rank ability across scenarios, immediately after first mentioning "aggregating all individual strengths using the geometric mean". We are willing to provide further clarification if the reviewer feels additional explanation is necessary. Please let us know if there are specific aspects you would like us to elaborate on.

---

> > ### Comment · Reviewer_jMjA · 2024-11-19
> >
> > > strength represents a player's overall skill or ranking within a group
> >
> > I think it would be clearer if `SE(s, a)` is written down precisely in terms of expectation and which distribution the expectation is taken over. Surely it depends on the distribution over opponent strategies and also depends on the probability of the player reaching state s?
> >
> > It is certainly true that a player at higher rank can lose to a player at a lower rank. The challenge, however, is that in this work an expected win-rate is used to derive an **action strength in each state that a player visit**. This does not seem to be well-grounded.
> >
> > Suppose we are playing a game of rock-paper-scissors and the human dataset is highly skewed, with 99 rocks, 1 paper, and 1 scissors. The paper player would have a much higher Elo rating (following the BT model) yet it loses catastrophically against the scissors player. Unfortunately the strength estimator should learn that taking the paper should win against scissors given the loss function that has been proposed.
> >
> > My comment about the reach probability hints a similar quality issue: in chess, suppose a highly rated player is great at most opening but is catastrophic at one opening $O$ which it plays rarely. The SE estimator would learn that every sequence of actions taken by this player following $O$ should be highly effective, since the player overall enjoys high ratings.
> >
> > Both factors could be detrimental to the quality of the learned SE network in fairly arbitrary ways which I think should be discussed more explicitly.

---

> > > ### Author Response · Authors · 2024-11-19
> > > **Response to Reviewer jMjA**
> > >
> > > We thank the reviewer for engaging in this discussion and sharing an interesting perspective.
> > >
> > > > … a game of rock-paper-scissors and the human dataset is highly skewed …
> > >
> > > Following the example of rock-paper-scissors, consider an online rock-paper-scissors platform based on the Elo rating system like FoxWeiqi and LiChess, where the strategy distribution among players is highly skewed – e.g., ~99% of players consistently choose rock, while <1% of players choose paper or scissors. In this scenario, a player consistently choosing paper would achieve the highest ranking, reflecting the effectiveness of strategies within this specific context.
> > >
> > > While theoretically all strategies should be equally ranked in rock-paper-scissors, a strength estimator predicting equal ranks for all strategies fails to predict player ranking on such a platform. Similarly, platforms like FoxWeiqi or LiChess do not provide theoretical rankings in Go or Chess but instead rank players based on their relative overall skill within that platform against other players.
> > >
> > > The above illustration highlights the goal of our paper: to train a strength estimator using human games from online platforms and evaluate its accuracy based on how well the predicted rankings align with the actual player rankings on those platforms, rather than theoretical rankings, as theoretical outcomes for complex games are infeasible beyond simple examples like rock-paper-scissors.
> > >
> > > > Unfortunately the strength estimator should learn that taking the paper should win against scissors given the loss function that has been proposed.
> > >
> > > We would like to clarify that the strength estimator is designed to learn that taking the paper **will rank higher than** scissors based on the proposed loss function, as the strength represents a player's overall **skill or ranking** within a group. As the reviewer mentioned, "It is certainly true that a player at higher rank can lose to a player at a lower rank." Therefore, a higher-ranked paper losing to a lower-ranked scissors is entirely acceptable within our framework.
> > >
> > > > My comment about the reach probability hints a similar quality issue: in chess, suppose a highly rated player is great at most opening but is catastrophic at one opening O which it plays rarely. The SE estimator would learn that every sequence of actions taken by this player following O should be highly effective, since the player overall enjoys high ratings.
> > >
> > > We are unsure if we fully understand the reviewer's concern. For example, on LiChess, if a highly rated player is catastrophic at a specific opening $O$, they would rarely play it, as frequent use could lower their rating. Consequently, the training dataset would contain only a small proportion of games involving opening $O$. Machine learning inherently optimizes for the majority of the data, meaning rarely played openings like $O$ would have minimal influence on training.
> > >
> > > We agree that the strength estimator might occasionally assign inaccurate strength scores, as the accuracy in Figure 2 does not reach 100%, and we do not claim that our estimator achieves perfection. Outliers, such as the example provided, are inevitable. However, we must emphasize that our SE and SE$_{\infty}$ models achieve a significant improvement in accuracy, increasing from 49% to over 80%, even in the occurrence of such scenarios.
> > >
> > > We hope the above explanation addresses your concern. Please let us know if we have misunderstood your question.
> > >
> > > > Both factors could be detrimental to the quality of the learned SE network in fairly arbitrary ways which I think should be discussed more explicitly.
> > >
> > > If the reviewer finds the above explanation satisfactory and has specific suggestions or additional aspects they would like us to elaborate on, we would be happy to incorporate further revisions into the paper. If not, we remain open to further discussion to address any concerns. Thank you again for your thoughtful follow-up and engagement.

---

> > > > ### Comment · Reviewer_jMjA · 2024-11-20
> > > >
> > > > > I think it would be clearer if SE(s, a) is written down precisely in terms of expectation and which distribution the expectation is taken over.
> > > >
> > > > I would like to ask again if it would be possible for the authors to write this down explicitly? I believe writing down the target that the SE network should approximate is at the core of the paper and would also help ground the discussion regarding the issues I brought up.
> > > >
> > > > > consider an online rock-paper-scissors platform based on the Elo rating system like FoxWeiqi and LiChess...
> > > >
> > > > My examples are thought experiments that are constructed specifically to make the point that deriving state-action level strength from aggregate strength estimates such as Elo scores can be flawed --- both due to intransitivity issues and state reach probability issues. Taking my contrived example then through a real-world thought continuation is not very productive.
> > > >
> > > > To clarify my reach probability concern: if a player A at Elo 2000 plays the opening $O$ 1% of the time but loses all the times at playing $O$. The current SE loss function would still assign high strength estimates from A's moves after $O$. In other words, player A's Elo scores accounts for the fact that $O$ is only played rarely, and on average, A remains a strong player. However, `SE(s, a)` with s subsequent to the opening O doesn't, as it predicts strength conditioned on which opening has been played.
> > > >
> > > > My overall point, which I don't believe is too controversial, is that the proposed SE is distilling coarse-grained statistics into fine-grained statistics, which can be arbitrarily incorrect at the finer granularity. This however seems to be a key assumption for the proposed method to be valid yet not discussed.
> > > >
> > > > An analogy of the assumption being made here is to say "housing in city A is more expensive than city B, therefore every house in city A is more expensive than city B". This assumption seems inherently difficult but not discussed.
> > > >
> > > > > we do not claim that our estimator achieves perfection..
> > > >
> > > > Certainly and I would not expect perfection for recommending acceptance of any paper. However, for a scientific publication (and not an engineered system) I believe these key assumptions should at least be discussed, or better, addressed by the method in principle.

---

> > > > > ### Author Response · Authors · 2024-11-20
> > > > > **Response to Reviewer jMjA**
> > > > >
> > > > > > I think it would be clearer if SE(s, a) is written down precisely in terms of expectation and which distribution the expectation is taken over.
> > > > >
> > > > > In our definition, the strength estimator SE(s, a) predicts a strength score $\lambda$, a positive value associated with the rank of the player who executed action $a$ in state $s$. $\lambda$ is a relative measure of comparative skill rather than an absolute or expected score.
> > > > >
> > > > > To connect this to expectations, consider two state-action pairs, $(s, a_1)$ and $(s, a_2)$, played by players with ranks $r_1$ and $r_2$, where $r_1$ is the higher rank. The strength estimator predicts $\lambda_1=SE(s, a_1)$ and $\lambda_2=SE(s, a_2)$ and aims to maximize the probability $P(\lambda_1 > \lambda_2)$. This probability is defined over the dataset $D$, which contains observed state-action pairs and ranks, and the expectation considers all such pairwise comparisons in $D$.
> > > > >
> > > > > > … the proposed SE is distilling coarse-grained statistics into fine-grained statistics, which can be arbitrarily incorrect at the finer granularity.
> > > > >
> > > > > We agree that a single state-action pair may not fully represent a player's overall skill. This is why we introduce the concept of *composite strength* in this paper, aggregating scores across multiple state-action pairs (using the geometric mean) to better capture a player's overall capabilities. This aggregation effectively shifts the focus from finer granularity to a coarser level, where the player's overall skill can be accurately estimated with **sufficient state-action pairs**. This is further supported by our experiment in Figure 2 (a), where using more games (state-action pairs) improves the strength estimator's accuracy.
> > > > >
> > > > >
> > > > > > … "housing in city A is more expensive than city B, therefore every house in city A is more expensive than city B" …
> > > > >
> > > > > If each house price in city A represents the strength score of a state-action pair played by player A, and similarly for city B, sampling just one house (state-action pair) from each city could lead to arbitrarily incorrect, as not every house in city A is more expensive than those in city B. However, by averaging prices across multiple houses from both cities, we can reliably conclude that housing in city A is more expensive overall. Similarly, aggregating strength scores across state-action pairs yields a more accurate estimation of a player's ranking.
> > > > >
> > > > > ---
> > > > >
> > > > > We hope the above explanation addresses the reviewer's concern regarding granularity. In our paper, we discuss the reasoning and motivation for using aggregation in the third and fourth paragraphs of subsection 3.1. However, if the reviewer feels the current description is insufficient, we are happy to provide a more detailed explanation in subsection 3.1 to further address this. Please let us know if you have any suggestions or additional concerns. Thank you again for your effort in reviewing our paper and engaging in this discussion.

---

> > > > > > ### Comment · Reviewer_jMjA · 2024-11-21
> > > > > >
> > > > > > Thank you for writing down the objective for the estimator $SE(s, a)$ a bit more specifically though here the objective remains contrastive. Nevertheless, it should be clear that what $SE(s, a)$ will converge to on a state action pair $(s, a)$ depends on
> > > > > >
> > > > > > 1. the probability of observing $s$ reached by a player of rank $r$ in the dataset $D$. This is what I was referring to as the reach probability. In some states, we may not observe any data from players of a certain rank. Some states may be reached by players of only one rank.
> > > > > >
> > > > > > 2. the probability of player of rank $r$ taking an action $a$ in $s$ in the dataset $D$. This again depends on what's present in the dataset $D$. Some actions may only have been taken in a state $s$ by player of one rank and no other rank.
> > > > > >
> > > > > > Since the learning objective is contrastive between ranks, it's unclear what $SE(s, a)$ should converge to in these situations. I appreciate that in practice your approach seem to perform better than prior works, but I cannot convince myself **why** it should work better as I don't know exactly what should $SE(s, a)$ converge to in all $(s, a)$ pairs.
> > > > > >
> > > > > > I thank the authors for engaging in the discussions but I think I'll keep my score.

---

> > > > > > > ### Author Response · Authors · 2024-11-21
> > > > > > > **Response to Reviewer jMjA**
> > > > > > >
> > > > > > > We appreciate the reviewer's response and clarification. We feel that, and we apologize if we are mistaken, the reviewer may have a fundamental misunderstanding regarding the core principles of supervised learning and the Bradley-Terry model.
> > > > > > >
> > > > > > > >  In some states, we may not observe any data from players of a certain rank. Some states may be reached by players of only one rank.
> > > > > > >
> > > > > > > >  Some actions may only have been taken in a state s by player of one rank and no other rank.
> > > > > > >
> > > > > > > Yes, both scenarios can occur, and this is a fundamental challenge in supervised learning: **the model is trained to fit the data available in the dataset while aiming to generalize effectively to unseen data**. In the examples you provided:
> > > > > > >
> > > > > > > - If some states are only reached by players of one rank => the SE model’s goal is to predict that rank for those states.
> > > > > > > - If certain actions are only taken in a state by players of one rank => the corresponding state-action pair is expected to predict that rank.
> > > > > > >
> > > > > > > We don't see any potential issue with the above scenarios. *May we ask if the reviewer's concern is whether SE might fail due to these rare states?*
> > > > > > >
> > > > > > > If so, consider the case of AlphaGo, where its policy network is trained on professional Go player games. In such games, there are inevitably some states where a professional player might make a mistake. However, the policy network is expected to predict that specific mistake of move in those states because it reflects the dataset's distribution. We kindly ask the reviewer: *in such cases, would the reviewer consider the policy network to be failing?*
> > > > > > >
> > > > > > >
> > > > > > > > Since the learning objective is contrastive between ranks, it's unclear what SE(s,a) should converge to in these situations.
> > > > > > >
> > > > > > > > I don't know exactly what should SE(s,a) converge to in all (s,a) pairs
> > > > > > >
> > > > > > > The Bradley-Terry model is inherently a **ranking problem**, where the values assigned represent **relative scores**, not absolute or expected values. The objective in ranking problems is to preserve the **relative relationships** between ranks rather than converging to a fixed or absolute value.
> > > > > > >
> > > > > > > To illustrate this, we kindly pose the following question to the reviewer: Suppose we are training a Bradley-Terry model to rank two players, A and B, where player A has a 60% win rate against player B. The goal is to find a function SE, where $SE(A) = \beta_A$ and $SE(B) = \beta_B$ such that the scores satisfy the Bradley-Terry model. In this scenario, *what exactly does the reviewer expect $SE(A)$ and $SE(B)$ converge to?*
> > > > > > >
> > > > > > > ---
> > > > > > >
> > > > > > > We sincerely hope the reviewer can address the above questions, as we believe they highlight the core disagreement and may reflect a misunderstanding of our work. We would greatly appreciate your feedback and suggestions for improvement.

---

> ### Author Response · Authors · 2024-11-19
> **Response to Reviewer jMjA (2/2)**
>
> > L238: "we sequentially perform the softmax loss...", typo? minimise the softmax loss?
>
> We have revised the sentence to "we sequentially minimize each softmax loss as defined by the equation 5." Thank you for pointing this out.
>
> > … optimise a rank-conditioned policy network π(⋅|s,r) with r the set of player ranks … SA-MCTS does not require ranked game-play data whereas the proposed methods do
>
> The idea of training a rank-conditioned policy network is interesting and was widely explored nearly a decade ago (Tian & Zhu, 2015). This approach was deprecated after AlphaGo due to its ineffectiveness in distinguishing ranks with input ranking features. Furthermore, as our goal is to establish a comprehensive strength system, it is unclear how such networks could contribute to strength estimation.
>
> Although our method requires ranked data for human-likeness, SA-MCTS, which does not rely on ranked data, produces less human-like moves. We view this as a trade-off for achieving human-likeness rather than a limitation. It is important to note that, as shown in subsection 4.4, our approach remains effective even with limited data.
>
> - Tian, Yuandong, and Yan Zhu. "Better computer go player with neural network and long-term prediction." arXiv preprint arXiv:1511.06410 (2015).
>
> ---
>
> We hope the above responses address the reviewer's concerns and strengthen the contributions of our strength system, which has significant potential for broad applications, similar to most real-world Elo rating systems, despite intransitivity issues. We kindly ask the reviewer to consider reevaluating our paper and are open to further discussions.

---

### Official Review · Reviewer_eKyZ · 2024-11-07

**Soundness:** 3
**Presentation:** 2
**Contribution:** 2
**Rating:** 5
**Confidence:** 3

**Summary:**

The authors introduce a strength estimation method that predicts rough player strength in go with only a few games. They use this as a method for running MCTS showing that it can be leveraged to approximate human-like play in go (and chess).

**Strengths:**

I think this is an interesting approach to modeling players in games and I like that the authors introduce a new metric for looking at player skill, but I am not fully convinced by the results.

# originality

This is a new approach to modeling humans and is breath of fresh air compared to the maximizing win rate approaches most other models use. I also like that this can be used for insight into players (as the authors note) and think this is a good new area of research.

# quality

The writing is OK, but I think the presentation needs another pass. The plots are difficult to read, I'm not clear how the confidence intervals are calculated, and the colours make telling lines apart difficult (figure 4 also has colour issues), consider some tables, log axis, larger lines/dots, smoothed lines, and fewer data points.

# clarity

I also found the paper not easy to follow the model labels aren't intuitive and are not defined in one place so it took some time to figure out what is meant by SE±1 vs SE_\inf±1 for example. I'm also not fully clear how the training and testing were done, which the lack of code exacerbates.

 # significance

I like the idea and this could be a significant new direction in the area, but the numerous issues with the paper limit it.

**Weaknesses:**

What is the training/testing split? I feel like I missed the section on how the dataset is constructed as I could only find a short discussion in section 4.1. I'm very concerned that there is some data leakage in how the experiments were run, specifically that the authors did not partition by players between training and testing. This would mean that the models can simply learn the preferred openings of each player and use that to predict skill. The numbers they get are about the same for a player identifier in chess [1].

The lack of baseline or consideration of alternative skill estimators I also find concerning as it makes evaluating the number much more difficult. Can the authors compare their results to the raw Elo estimates given by the 100/15 games (using the opponents known Elo) or better yet compare to some simple classifier.

I am also concerned by lack of code release as that is an important part of the final paper.


[1] McIlroy-Young, Reid, et al. "Detecting individual decision-making style: Exploring behavioral stylometry in chess." Advances in Neural Information Processing Systems 34 (2021): 24482-24497.

**Questions:**

+ Why did the authors not compare their results to any of the models of human-like play in chess or go?
+ I'm unclear on the reasoning for the claim on line 534 "However, this issue could potentially be addressed by using all models trained by AlphaZero, which", how would that work? Elo is a relative measure (expected winrate vs the community), how would AlphaZero's Elo be relevant for humans without human games?

---

> ### Author Response · Authors · 2024-11-19
> **Response to Reviewer eKyZ (1/2)**
>
> We thank the reviewer for finding our work interesting and providing insightful feedback. We would like to clarify the concerns raised, particularly regarding the issue of data leakage.
>
> > What is the training/testing split? … partition by players between training and testing … simply learn the preferred openings of each player and use that to predict skill …
>
> We would like to clarify a potential misunderstanding regarding the objective of our work compared to [1]. *Playing strength* differs fundamentally from *playing style* or *player-specific preferences*. Strength reflects the overall skill level of a group of players with similar rankings, regardless of individual styles. For instance, two Go players of the same rank may use different strategies (e.g., offensive vs. defensive), but our approach classifies them into the same rank, while [1] distinguishes players based on style. On the other hand, if a player improves from 1 dan to 2 dan, our strength estimator classifies the player's games by rank, classifying them into 1 dan and 2 dan accordingly, while [1] classifies all games under the same individual, ignoring rank progression.
>
> Consequently, splitting training/testing datasets by players is unnecessary, as each strength corresponds to a specific rank involving many players with diverse styles. For instance, Figure 3 shows that strength scores are nearly identical during the opening, likely because early moves in Go have limited variety, and weaker players can easily imitate stronger ones. This confirms that the models cannot rely on play-specific opening preferences to predict strength, unlike [1].
>
> In our experiment, training and testing datasets contain entirely distinct games, though overlapping players may be present, which is acceptable. We apologize for the unclear description in the initial version and have revised the paper to explicitly describe datasets for both Go and chess in subsections 4.1 and 4.5. Thank you for pointing this out!
>
> [1] McIlroy-Young, Reid, et al. "Detecting individual decision-making style: Exploring behavioral stylometry in chess." Advances in Neural Information Processing Systems 34 (2021): 24482-24497.
>
> > The lack of baseline or consideration … compare their results to the raw Elo estimates given by the 100/15 games …
>
> Our paper already includes comparable baseline models (Moudˇrík & Neruda, 2016) for strength estimation, specifically $SL_{sum}$ and $SL_{vote}$, which provide direct comparisons for evaluating the accuracy of predicting playing strength from game records. Regarding the comparison to raw Elo estimates over 100/15 games, we would greatly appreciate more detailed guidance or references to related papers from the reviewer on how to conduct this experiment. We are more than willing to perform additional experiments to address your concerns.
>
> > I am also concerned by lack of code release …
>
> We understand the importance of releasing code. As stated in our REPRODUCIBILITY STATEMENT: "The source code, along with a README file containing instructions, will be released to ensure reproducibility once this paper is accepted." However, if the reviewer finds it essential to examine the code during the review process, we are happy to upload it to an anonymous repository for review.
>
> > compare their results to any of the models of human-like play …
>
> The primary focus of our paper is on a strength system that (a) estimates player strength from their historical game records and (b) adjusts strength to provide suitable levels, with the additional benefit of a more human-like playing style. Given these goals and objectives, we compare our strength estimator with two $SL$ networks and evaluate SE-MCTS against MCTS and SA-MCTS for strength adjustment.
>
> We understand that several studies in chess, such as  Maia (McIlroy-Young et al., 2020) and Maia-2 (Tang et al., 2024), focus on human-like play. However, these works mainly aim to achieve high accuracy in predicting human moves, without estimating the strength or verifying whether the actual playing strength aligns with specific rankings. Given the distinct goals and objectives of our work, we believe a direct comparison may not be fully applicable.
> * McIlroy-Young, Reid, et al. "Aligning superhuman AI with human behavior: Chess as a model system." Proceedings of the 26th ACM SIGKDD International Conference on Knowledge Discovery & Data Mining. 2020.
> * Tang, Zhenwei, et al. "Maia-2: A Unified Model for Human-AI Alignment in Chess." arXiv preprint arXiv:2409.20553 (2024).

---

> ### Author Response · Authors · 2024-11-19
> **Response to Reviewer eKyZ (2/2)**
>
> > I'm unclear on the reasoning for the claim on line 534 … how would AlphaZero's Elo be relevant for humans without human games?
>
> Yes, Elo is a relative measure, and our strength estimator similarly learns relative strength scores for state-action pairs, with higher-ranking players associated with higher scores. It does not provide absolute rankings or Elo ratings. Therefore, **human games are necessary during testing** to calibrate these scores with specific rankings. For instance, as shown in Figure 3, a value around 1 to 2 is aligned to a rank like 9 dan using human data. On the other hand, **to reduce reliance on human games during training**, especially for niche games, we propose using AlphaZero self-play games to learn strength scores, which can be used to predict human ranks.
>
> > … plots are difficult to read … how the confidence intervals are calculated
>
> We describe the method of evaluation in the third paragraph of subsection 4.2. Specifically, we randomly select $N$ games (from 1 to 100) for each $r_i$ from the query dataset. Since a single sample can only result in an accuracy of 1 (correct prediction) or 0 (incorrect prediction), we repeat this process 500 times to ensure a stable estimation. For example, when $N=5$, we randomly sample five games from the query dataset and check whether the strength estimator predicts the correct rank based on these games. This process is repeated 500 times, and the resulting prediction distribution is used to calculate the confidence intervals.
>
> Regarding the plots, we have revised the figures to improve readability by using bold and smoothed lines and lighter colors. Please let us know if the reviewer finds the revision clearer or has additional suggestions.
>
> ---
>
> We hope the above clarification resolves the reviewer's potential confusion between our work and [1], and demonstrates that our experimental results are robust and reliable. We remain open to further discussions. Thank you again for your thoughtful review.

---

> ### Author Response · Authors · 2024-11-25
> **Follow Up**
>
> Dear Reviewer eKyZ,
>
> Thank you for your review and valuable feedback. As the discussion phase is ending, we would greatly appreciate it if you could review our rebuttal, particularly regarding the training and testing dataset. We believe our experimental results are robust and hope the reviewer will consider a more favorable recommendation after reviewing our rebuttal.
>
> We are more than willing to provide additional explanations or experiments within the remaining discussion period. Thank you again for your time and consideration.
>
> Sincerely,
>
> Authors

---

> > ### Comment · Reviewer_eKyZ · 2024-11-25
> >
> > I apologize for the late response. I have read your response and the other reviewers discussions. I continue to be skeptical of the results and am not convinced you are measuring what you claim to be. I maintain my score.
> >
> > I still do not think your training/testing split is correct. You say that _a specific rank involving many players with diverse styles_, but do not provide numbers. On Lichess there are under 2000 players with ratings above 2500 for blitz, that's 5 dan and higher. As you mentioned it is easy to learn the style of a player from only a few games, thus a model that learns a few different styles is also plausible. You need to test for this, your assertions may be correct, but I suspect accuracy will drop significantly once you test on new set of players.
> >
> > Regarding using the normal Elo formula. Here it is from Wikipedia:
> >
> > $R_\mathsf{A}' = R_\mathsf{A} + K \cdot (S_\mathsf{A} - E_\mathsf{A})$
> >
> > Usually players start at 1500, then it updates after each game. You can run this if you know the opponent's Elo.
> >
> > I asked you to explain the reasoning behind your statement on line 534, you simply rephrased it _On the other hand, to reduce reliance on human games during training, especially for niche games, we propose using AlphaZero self-play games to learn strength scores, which can be used to predict human ranks._ Please explain how self-play can help on human games, also keep in mind that comparing between human games on different servers is very non-trivial, and as reviewer jMjA discusses is not stable even on a single server.

---

> > > ### Author Response · Authors · 2024-11-26
> > >
> > > Thank you for your response. We address the questions below.
> > >
> > > > a specific rank involving many players with diverse styles, but do not provide numbers
> > >
> > > We apologize for not including the numbers. We provide the detailed number of players for each rank in the training dataset below. Each rank includes thousands of players, making it challenging for a model to learn all player's styles.
> > >
> > > - Go (Training dataset)
> > > |Rank|Number of Players|Number of Games|
> > > |-|-|-|
> > > |3-5 kyu|16,900|45,000|
> > > |1-2 kyu|18,325|45,000|
> > > |1 dan|27,522|45,000|
> > > |2 dan|25,459|45,000|
> > > |3 dan|28,045|45,000|
> > > |4 dan|27,658|45,000|
> > > |5 dan|24,321|45,000|
> > > |6 dan|19,642|45,000|
> > > |7 dan|15,527|45,000|
> > > |8 dan|12,191|45,000|
> > > |9 dan|7,520|45,000|
> > > |Total|159,262|495,000|
> > >
> > > - chess  (Training dataset)
> > > |Elo|Number of Players|Number of Games|
> > > |-|-|-|
> > > |1000-1199|158,176|240,000|
> > > |1200-1399|168,704|240,000|
> > > |1400-1599|171,428|240,000|
> > > |1600-1799|159,399|240,000|
> > > |1800-1999|133,565|240,000|
> > > |2000-2199|92,348|240,000|
> > > |2200-2399|48,631|240,000|
> > > |2400-2599|18,863|240,000|
> > > |Total|764,720|1,920,000|
> > >
> > > > You need to test for this, your assertions may be correct …
> > >
> > > Certainly! To address the reviewer's concern regarding the overlapping, we have conducted an experiment using entirely different players for training and testing datasets. The number of players and games in testing dataset for each rank is as follows:
> > >
> > > - Go  (Testing dataset with non-overlapping players)
> > > |Rank|Number of Players|Number of Games|
> > > |-|-|-|
> > > |3-5 kyu|1,216|1,000|
> > > |1-2 kyu|1,157|1,000|
> > > |1 dan|1,123|1,000|
> > > |2 dan|1,085|1,000|
> > > |3 dan|1,109|1,000|
> > > |4 dan|1,064|1,000|
> > > |5 dan|1,086|1,000|
> > > |6 dan|1,062|1,000|
> > > |7 dan|1,002|1,000|
> > > |8 dan|893|1,000|
> > > |9 dan|206|1,000|
> > > |Total|10,114|11,000|
> > >
> > > - chess (Testing dataset with non-overlapping players)
> > > |Elo|Number of Players|Number of Games|
> > > |-|-|-|
> > > |1000-1199|2,366|1,220|
> > > |1200-1399|2,362|1,220|
> > > |1400-1599|2,350|1,220|
> > > |1600-1799|2,344|1,220|
> > > |1800-1999|2,316|1,220|
> > > |2000-2199|2,166|1,220|
> > > |2200-2399|1,796|1,220|
> > > |2400-2599|1,092|1,220|
> > > |Total|16,502|9,760|
> > >
> > > The results below show that accuracy remains consistent in both Go and chess, regardless of whether overlapping or non-overlapping players are used. This indicates that our model focuses on learning the strength rather than relying on player styles.
> > >
> > > - Go
> > > |Games|Testing data (paper)|Testing dataset (No overlapping players)|
> > > |-|-|-|
> > > |1|36.91%±4.23%|33.82%±4.15%|
> > > |10|73.27%±3.88%|70.49%±4.00%|
> > > |20|83.82%±3.23%|81.51%±3.40%|
> > > |30|89.36%±2.70%|86.53%±2.99%|
> > > |40|91.18%±2.48%|89.36%±2.70%|
> > > |50|94.27%±2.04%|91.29%±2.38%|
> > > |60|94.68%±1.97%|92.71%±2.12%|
> > > |70|95.77%±1.76%|93.69%±1.92%|
> > > |80|96.23%±1.67%|94.05%±1.81%|
> > > |90|96.86%±1.53%|94.73%±1.68%|
> > > |100|97.44%±1.39%|95.16%±1.57%|
> > >
> > > - chess
> > > |Games|Testing data (paper)|Testing dataset (No overlapping players)|
> > > |-|-|-|
> > > |1|31.37%±4.07%|31.31%±4.08%|
> > > |10|63.31%±4.22%|61.81%±4.27%|
> > > |20|74.62%±3.81%|75.38%±3.78%|
> > > |30|82.75%±3.31%|82.88%±3.31%|
> > > |40|85.06%±3.12%|88.12%±2.84%|
> > > |50|89.38%±2.52%|91.00%±2.52%|
> > > |60|90.56%±2.28%|92.75%±2.28%|
> > > |70|92.50%±2.18%|93.44%±2.18%|
> > > |80|92.88%±1.82%|95.50%±1.82%|
> > > |90|94.00%±1.71%|96.06%±1.71%|
> > > |100|94.50%±1.39%|97.44%±1.39%|
> > >
> > > > explain how self-play can help on human games
> > >
> > > Evidence suggests that AI strength may correlate with human strengths. For example, [2] demonstrated that an AlphaZero-like program trained solely on self-play games could correspond to human Go kyu/dan rankings. This indicates the *potential* feasibility of using AI self-play games to help on human games for training strength estimators, which we highlight as a possible direction for *future work*.
> > >
> > > We agree with the reviewer that the AI self-play games may not directly reflect human player strength relationships. This is why we carefully use the term *could potentially* in our paper and do not claim this as definitive, as verifying it requires non-trivial effort.
> > >
> > > We are happy to include the above explanation in the discussion section if the reviewer finds it necessary.
> > >
> > > [2] Liu, An-Jen, et al. "Strength adjustment and assessment for MCTS-based programs." IEEE Computational Intelligence Magazine 15.3 (2020): 60-73.
> > >
> > > > comparing between human games on different servers is very non-trivial
> > >
> > > Yes, we agree. All of our experiments are conducted on the same platform (FoxWeiqi and LiChess) for both training and testing data.
> > >
> > > ---
> > >
> > > We hope the above experiments address your concerns, and are happy to answer any further questions or provide additional experiments.

---

### Author Response · Authors · 2024-11-19
**Summary of Revision**

**Update** (suggested by Reviewer KBHz):

We have included the strength estimator for chess in Figure 7 and the following in Appendix C:
* A table of exact values of $z$ and how these values were determined.
* A round-robin table for using different baseline models.
---

Dear all reviewers,

We sincerely appreciate the reviewers' thoughtful comments and constructive feedback. We have uploaded a revised version based on the suggestions. To make it easier to identify the changes, all revisions are highlighted in red text. Below is a summary of the updates:

### **Revisions**
* Add the relation and distinction between strength estimation and learning to rank in subsection 2.4, on page 3. (Reviewer KBHz)
* Revise the phrase "sequentially perform the softmax loss" to "sequentially minimize each softmax loss" in subsection 3.2, on page 5. (Reviewer jMjA)
* Provide detailed information on training/testing datasets for both Go and chess experiments in subsections 4.1 and 4.5, on pages 6 and 9. (Reviewer eKyZ)
* Clarify the specific model used in Figure 3 within the figure caption, on page 8. (Reviewer KBHz)
* Add a citation for SA-MCTS in subsection 4.3, on page 8. (Reviewer KBHz)
* Include the strength estimator for chess in Figure 6, 7, and Table 2, along with revised sentences in subsection 4.5, on page 10. (Reviewer KBHz)
* Add a discussion on addressing intransitivity issues in the Bradley-Terry model as a direction for future research direction, on page 10. (Reviewer jMjA)
* A table of exact values of $z$ and how these values were determined. (Reviewer KBHz)
* A round-robin table for using different baseline models. (Reviewer KBHz)

Please let us know if you have any additional suggestions. We hope this revision contributes to a more favorable recommendation for our paper.


### **Figure Improvements**
* Improve the readability of Figures 2, 5, and 6 (as well as several figures in the appendix) by using bold and smoothed lines. All results remain consistent with the initial version. (Reviewer eKyZ)
* Improve the readability of Figures 4 and 7 by using lighter colors and reducing space sizes. All results remain consistent with the initial version. (Reviewer eKyZ)

We hope these revisions improve the clarity and quality of the paper. If there is anything further we can do to address your concerns, please let us know. We are willing to engage in further discussions or conduct additional experiments as needed during the rebuttal period. Thank you for your time and effort.

---

### Meta-Review · Area_Chair_cChN · 2024-12-23

**Metareview:**

An MCTS version that plays at a particular strength, and a new strength estimator to identify said strength. There is novelty and the results are good. Two reviewers recommend acceptance, and the most negative reviewer appears to have misunderstood aspects of the problem (and potentially of the proposed solution). On balance, I think this paper can be accepted.

**Additional Comments On Reviewer Discussion:**

The authors went out of the way to provide good answers to reviewers, run new analyses etc.

---

### Decision · Program_Chairs · 2025-01-22

Accept (Poster)